# Global patterns and climatic controls of forest structural complexity

Martin Ehbrecht[1✉], Dominik Seidel [1], Peter Annighöfer [2], Holger Kreft[3,4], Michael Köhler[5], Delphine Clara Zemp[3], Klaus Puettmann[6], Reuben Nilus[7], Fred Babweteera[8,9], Katharina Willim [1], Melissa Stiers[1], Daniel Soto[10], Hans Juergen Boehmer [11,12], Nicholas Fisichelli[13], Michael Burnett [14,15], Glenn Juday[16], Scott L. Stephens[17] & Christian Ammer [1,4]

The complexity of forest structures plays a crucial role in regulating forest ecosystem functions and strongly influences biodiversity. Yet, knowledge of the global patterns and determinants of forest structural complexity remains scarce. Using a stand structural complexity index based on terrestrial laser scanning, we quantify the structural complexity of boreal, temperate, subtropical and tropical primary forests. We find that the global variation of forest structural complexity is largely explained by annual precipitation and precipitation seasonality ($R^2 = 0.89$). Using the structural complexity of primary forests as benchmark, we model the potential structural complexity across biomes and present a global map of the potential structural complexity of the earth´s forest ecoregions. Our analyses reveal distinct latitudinal patterns of forest structure and show that hotspots of high structural complexity coincide with hotspots of plant diversity. Considering the mechanistic underpinnings of forest structural complexity, our results suggest spatially contrasting changes of forest structure with climate change within and across biomes.

[1] Silviculture and Forest Ecology of the Temperate Zones, University of Göttingen, Büsgenweg 1, 37077 Göttingen, Germany. [2] Forest and Agroforest Systems, Technical University of Munich (TUM), Hans-Carl-von-Carlowitz-Platz 2, 85354 Freising, Germany. [3] Biodiversity, Macroecology and Biogeography, University of Göttingen, Büsgenweg 1, 37077 Göttingen, Germany. [4] Centre of Biodiversity and Sustainable Land Use (CBL), University of Göttingen, Büsgenweg 1, 37077 Göttingen, Germany. [5] Northwest German Forest Research Institute, Grätzelstr. 2, 37079 Göttingen, Germany. [6] Department of Forest Ecosystems and Society, Oregon State University, Corvallis, OR 97331, USA. [7] Forest Research Centre, Sabah Forestry Department, P.O. Box 1407, 90715 Sandakan, Malaysia. [8] Budongo Conservation Field Station, P.O. Box 362Masindi, Uganda. [9] Department of Forestry, Biodiversity and Tourism, Makerere University, P.O. Box, 7062 Kampala, Uganda. [10] Departmento de Recursos Naurales y Tecnología, Universidad de Aysén, Obispo Vielmo 62, Coyhaique, Chile. [11] School of Geography, Earth Science, and Environment, University of the South Pacific, Laucala Bay, Suva, Fiji. [12] Institute of Geography, University of Jena, Löbdergraben 32, 07743 Jena, Germany. [13] Schoodic Institute at Acadia National Park, P.O. Box 277Winter Harbor, ME 04693, USA. [14] Earth Systems Program, Stanford University, 473 Via Ortega, Stanford, CA 94305, USA. [15] The Nature Conservancy, 67-1197 Mamalahoa Hwy.P. O. Box 1056Kamuela, HI 96743, USA. [16] Department of Natural Resources and Environment, and Institute of Agriculture, Natural Resources and Extension, University of Alaska Fairbanks, P.O. Box 7566180Fairbanks, AK 99775, USA. [17] Department of Environmental Science, Policy, and Management, University of California, 130 Mulford Hall, Berkeley, CA 94720, USA. ✉email: martin.ehbrecht@forst.uni-goettingen.de

Climate change will alter the structure and functioning of boreal, temperate and tropical forest ecosystems with contrasting, yet unclear impacts on biodiversity and ecosystem functions across biomes[1–3]. Responses of forest biodiversity and ecosystem functions to climate change are strongly linked to changes in forest structural complexity[4–7]. Consequently, understanding the impacts of climate change on forest biodiversity and ecosystem functions requires an in-depth understanding of the climatic controls on forest structural complexity[8]. Climate shapes forest compositional and functional diversity, which are important determinants of forest structural complexity[9,10]. However, it remains unclear how relationships between climate and compositional and functional diversity translate into global patterns of forest structural complexity. Understanding the climatic determinants and global patterns of forest structural complexity could provide an urgently needed basis to better predict how biodiversity and ecosystem functions will respond to climate change.

Forest structural complexity aims to quantify the distribution of trees and their canopies in three-dimensional space, thus expanding beyond summarizing forest structure in structural attributes such as biomass, leaf area or canopy height[11,12]. At the stand level, greater structural complexity manifests itself in a higher diversity of tree sizes and crown morphologies[11], resulting in multi-layered and more densely-packed canopies and a greater connectedness of individual tree canopies[13] (Fig. 1). Forest structural complexity can thus be defined by the degree of heterogeneity in biomass distribution in three-dimensional space and depends on the spatial patterns and efficiency of canopy space occupation (sensu[14], Supplementary Fig. 3). First used to address key ecological questions such as the habitat heterogeneity-biodiversity relationship[15], measures of forest structural complexity have recently proven useful for understanding interactions between three-dimensional forest structure, biodiversity, and ecosystem functions[4,13,16]. The increased availability of airborne and terrestrial LiDAR (Light Detection and Ranging) technologies for forest ecology applications, which provides an opportunity to quantify the three-dimensional nature of forest structure (sensu[17]), has triggered the development of new methodologies and metrics to quantify forest structural complexity[18]. Measures of structural complexity have proven to be strong predictors of net primary productivity[6,7], because important drivers of forest growth, such as occupied canopy space[19,20], connectedness of tree canopies[13], and thereby light absorption[21,22], are accounted for in structural complexity metrics.

Tree species composition, complementarity in crown architectures and tree size diversity (vertical stratification) together determine the spatial patterns and efficiency of canopy space occupation and thus forest structural complexity[23] (Fig. 1). For example, recent studies have shown that tree species diversity positively affects structural complexity[23–25], as higher tree species diversity may result in complementary canopy space occupation due to contrasting crown architectures, thereby increasing canopy packing and complexity[26,27] (niche complementarity). However, the co-existence and growth of different tree species, tree sizes and morphologies in different canopy layers depends on their physiological traits with respect to shade tolerance, crown plasticity and the ability to acquire belowground resources under stress from competition[28]. Thus, forest structural complexity is constrained by functional diversity and the range of plant functional strategies[29].

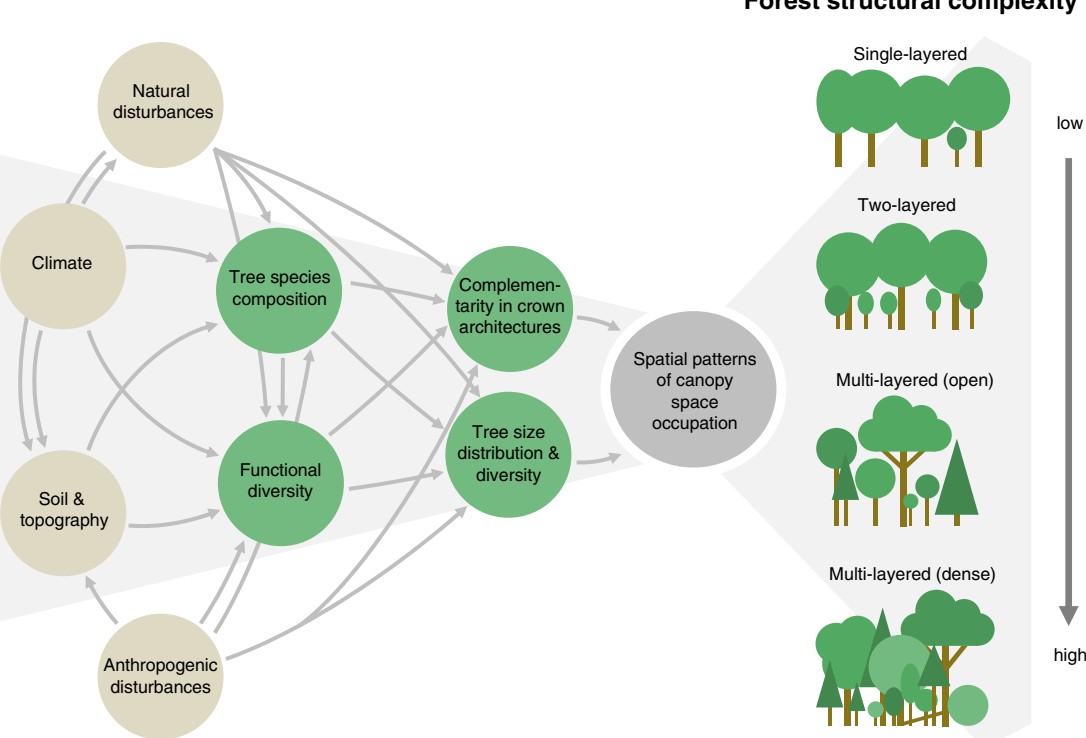

**Fig. 1 Conceptual figure outlining the abiotic and biotic controls on forest structural complexity.** Forest structural complexity increases with greater diversity of tree sizes and complementarity in crown architectures.

Forest compositional and functional diversity are strongly controlled by climate, with more humid and warmer climates supporting a wider spectrum of plant functional strategies (physiological tolerance hypothesis)[10]. This suggests that climate shapes forest structural complexity through its controls on forest compositional and functional diversity. How the climatic controls on forest compositional and functional diversity translate into global patterns of forest structural complexity remains, however, largely unexplored. To unravel the future effects of climate change on forest ecosystem functions and biodiversity[30,31], deeper insights into how climate shapes forest structural complexity and its global patterns are urgently needed, because both, ecosystem functions and biodiversity are strongly influenced by forest structure[4,6,7].

Here, we aim at contributing to a better understanding of the global variation and the climatic drivers of forest structural complexity across biomes, to map its global patterns and to estimate its responses to climate change. An in-depth understanding of climatic controls on forest structural complexity can only be gained by investigating primary forests with negligible anthropogenic and natural disturbances on forest structure (Fig. 1). Despite recent advances in satellite and airborne laser scanning[32], our knowledge of global patterns of forest structural complexity, and how these relate to climate, remains largely incomplete. Therefore, we conduct an extensive global field campaign in undisturbed, primary boreal forests, temperate broadleaf and temperate conifer forests, tropical moist broadleaf forests, as well as subtropical tree savannas. We quantify their three-dimensional structure and complexity using the well-established, terrestrial LiDAR-based stand structural complexity index SSCI[11]. We additionally measure canopy height, canopy openness, and basal area (as proxy for above-ground biomass) as major attributes of forest structure and link all those forest structural parameters to climatic variables.

We hypothesize that the global variation of forest structural complexity is mainly determined by the climatic factors that control compositional and functional diversity, namely light availability during the growing season (solar radiation (kJ m$^{-2}$ day$^{-1}$), mean temperature during the growing season (°C), and water availability (mean annual precipitation (mm), precipitation seasonality (coefficient of variation (%)), and mean annual precipitation minus potential evapotranspiration (mm)). We use globally modeled climate data (see ref. [33]) to test relationships between those climate variables and forest structural complexity, canopy height, basal area and canopy openness. Furthermore, we include edaphic factors in our analysis, namely soil water holding capacity (field capacity in cm$^3$ cm$^{-3}$), soil nitrogen content (g kg$^{-1}$) and cation exchange capacity (mmol (c) kg$^{-1}$) to control for probable soil-related effects. We find that the global variation of forest structural complexity is largely explained by annual precipitation and precipitation seasonality. Using the structural complexity of primary forests as benchmark, we provide a global estimate of the potential structural complexity across biomes and realms. The resulting map can provide a reference for forest management and restoration, as well as to better determine the structural intactness of the world's forests.

## Results

The structural complexity of primary forests, quantified by the stand structural complexity index, SSCI, strongly correlated with annual precipitation, precipitation seasonality, the water balance and soil water holding capacity (field capacity) across biomes (Fig. 2). We did not find a significant relationship with mean annual temperature (MAT), mean growing season temperature or cation exchange capacity as single predictors. Light availability, as measured by solar radiation during the growing season, and soil nitrogen were correlated with SSCI, but explained less variation than water availability-related variables. We then tested all possible combinations of explanatory variables in multiple regression models (Table 1, only models where each explanatory variable was significant at $p < 0.05$ are shown).

To avoid collinearity, we only combined variables where intercorrelation did not exceed a threshold of $r < |0.7|$[34] (see Supplementary Fig. 5). A multiple linear regression model of mean annual precipitation (MAP) and precipitation seasonality (coefficient of variation (%)) explained 89.4% of variation in structural complexity across biomes (see Table 1 and Supplementary Fig. 6) and performed better than any other model ($\Delta AIC_c = 9.77$), which was further confirmed by an automated model selection algorithm (MuMln R-package v1.43.17). Mean annual temperature and growing season temperature had a significant effect on SSCI in combination with water balance (MAP–PET), but explained less variation and had a higher root mean square error than the ´best´ model. Model residuals were not spatially auto-correlated (observed Moran´s I = 0.006, $p = 0.19$, spdep R-package v.1.1-3).

The robustness of the 'best' model, with only mean annual precipitation and seasonality as predictors, was evaluated by a leave-one-out-cross-validation approach that predicted the structural complexity of excluded sites with a RMSE of 0.71 and an $R^2$ of 0.86. Moreover, excluding entire biomes from the model did not reduce its explanatory power, except for the exclusion of Subtropical Savannas and Woodlands ($R^2 = 0.82$, Table 2). We did not find significant relationships between climate and soil variables and canopy height or basal area. Canopy openness, however, exponentially decreased with increasing mean annual precipitation and increased with seasonality (see Supplementary Fig. 7).

Using globally available climate data for the period 1971−2000 from the WorldClim2 database[33] and the structure-climate model from our analysis, we predicted and mapped the potential structural complexity (SSCI$_{pot}$) for all ecoregions that were classified as forest or woodland according to Olson et al.[35] at 30 arcsecond resolution (Fig. 4a). SSCI$_{pot}$ quantifies the structural complexity that could potentially develop at a given site without anthropogenic disturbance and reflects the potential climate-defined climax of forest structural complexity. To avoid model extrapolation, we only made predictions for biomes that were included in our sample. Consequently, tropical and subtropical dry broadleaf and conifer forests, mangroves, and Mediterranean forests and woodlands were excluded.

On a global scale, SSCI$_{pot}$ decreases from (sub-) tropical moist broadleaf forests (mean SSCI$_{pot} = 6.79$) to temperate broadleaf (mean SSCI$_{pot} = 5.75$), to temperate conifer (mean SSCI$_{pot} = 5.15$), to boreal forests (mean SSCI$_{pot} = 4.99$) and finally to (sub-) tropical savannas and woodlands (mean SSCI$_{pot} = 4.54$) (Fig. 3a). However, SSCI$_{pot}$ varied largely within biomes, especially in the tropical and subtropical moist broadleaf forest biome, which covers the broadest climatic range (see Supplementary Fig. 1). Following a distinct latitudinal pattern (Fig. 3b), SSCI$_{pot}$ peaks at the equator, decreases sharply towards the Tropics of Cancer and Capricorn, and increases again towards the mid latitudes, peaking at around 40° north and south in the temperate zones, after which it decreases again towards the boreal zone in the northern hemisphere.

Hotspots of very high potential structural complexity (SSCI$_{pot} \geq 9$) were found in ecoregions of Australasian, Indomalayan and Neotropical moist broadleaf forests, including the Napo and Choco-Darien moist forests in Western Amazonia, Borneo, and Sumatra lowland rainforests, and New Guinean lowland rainforests in insular south-east Asia (Fig. 4a). In the temperate zones, hotspots of high SSCI$_{pot}$ were found in temperate rainforest ecoregions such as the Valdivian temperate forest in southern America, the Northern Pacific Alaskan coastal forest in

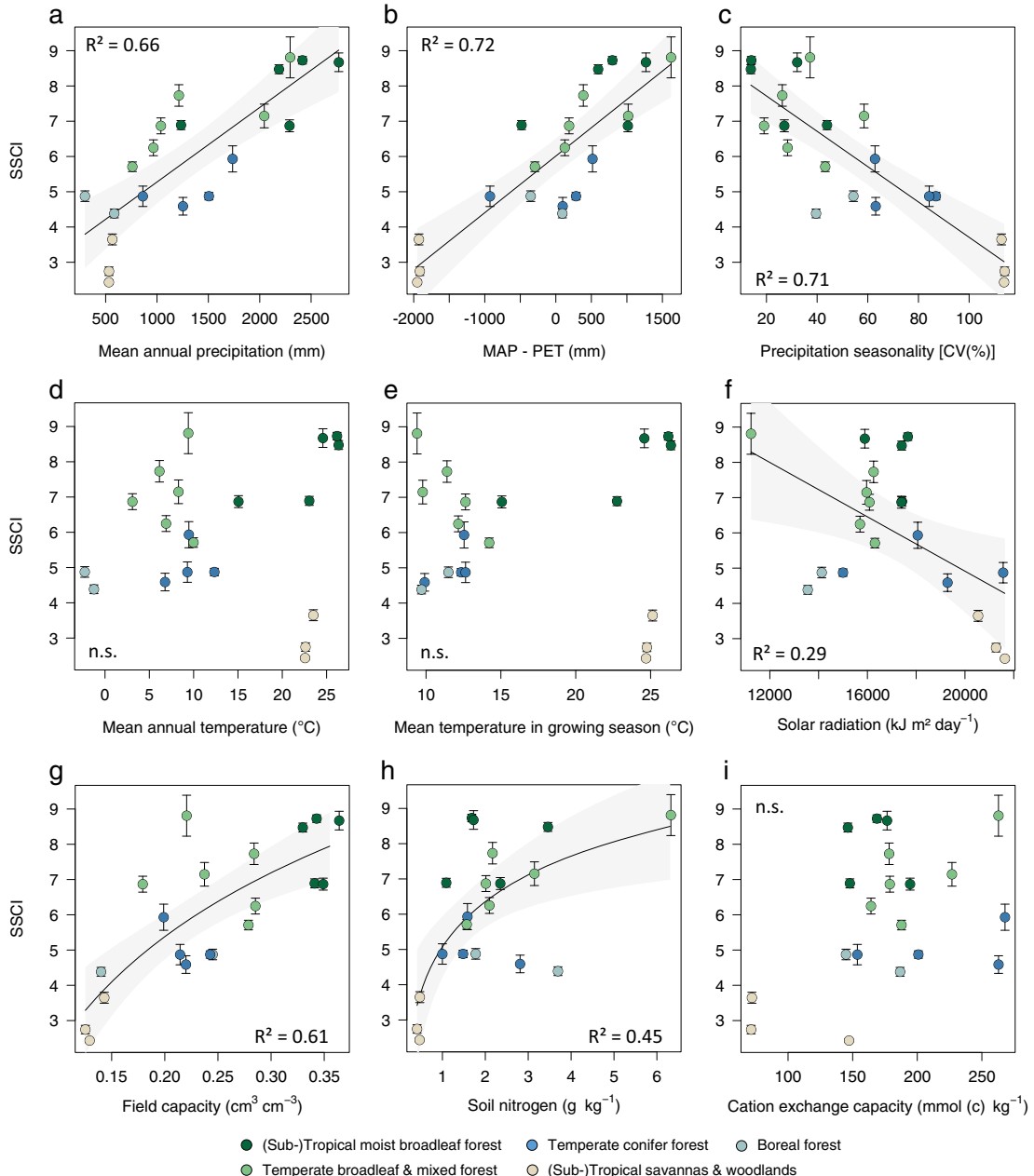

**Fig. 2 Relationships of forest structural complexity (SSCI) with climatic and edaphic factors.** Linear regression was used to model relationships between primary forest structural complexity, as quantified by the stand structural complexity index (SSCI) and **a** mean annual precipitation (mm), **b** water balance (MAP–PET in mm), **c** precipitation seasonality (coefficient of variation in %), **d** mean annual temperature (°C), **e** mean temperature during growing season (°C), **f** solar radiation (kJ m² ha⁻¹), **g** soil water holding capacity (field capacity in cm³ cm⁻³), **h** soil nitrogen (g kg⁻¹), **i** cation exchange capacity (mmol (c) kg⁻¹). Data points represent mean SSCI values for each site ($n = 20$ sites). Error bars indicate the standard error of the mean SSCI per site. Number of plots per site are shown in Table 3. Shaded envelopes represent the 95% confidence interval of the regression lines.

northern America and the Tasmanian temperate rainforest in Australia. The (sub-) tropical grasslands, savannas and shrubland biome includes woodland ecoregions that characterize lower end of global $SSCI_{pot}$, including Angolan Mopane woodlands, Zambezian Baikiaea, and Miombo woodlands in southern and south-eastern Africa.

## Discussion
We report results from an extensive global field campaign, modeling and scaling up the structural complexity of boreal,

temperate, subtropical and tropical primary forests based on terrestrial LiDAR data. Forest structural complexity was strongly correlated with water availability across all evaluated biomes. The best performing model, which leveraged mean annual precipitation and its seasonality as explanatory variables, explained 89.4% of variation in the forest structural complexity index (Table 1).

Climate-structure relationships are most likely controlled by relationships between climate and the functional traits or structural attributes that interact to create complex three-dimensional forest structures. The spatial patterns and efficiency of canopy space occupation beneath the canopy of the tallest trees are

**Table 1 Coefficient of determination ($R^2$), Akaike Information Criterion (AICc), difference in AICc value between the respective model and the ´best´ model (ΔAICc), root mean square error (RMSE) and Moran´s I of linear (lm) and linear mixed effects models (lme) used to predict stand structural complexity index (SSCI) based on climate and soil variables.**

| Model | Predictor variables | $r^2$ | AICc | Δ AICc | RMSE | Moran´s I |
|---|---|---|---|---|---|---|
| lm | MAP + prec. seasonality | 0.89 | 48.02 | 0.00 | 0.62 | −0.006 |
| lm | MAP + temp. during growing season | 0.83 | 57.79 | 9.77 | 0.79 | −0.051 |
| lm | MAP-PET + MAT | 0.81 | 60.22 | 12.20 | 0.84 | −0.087 |
| lm | Field capacity + nitrogen + MAP | 0.83 | 61.28 | 13.26 | 0.78 | −0.052 |
| lm | Prec. seasonality | 0.77 | 63.19 | 15.17 | 0.89 | −0.116 |
| lm | MAP + field capacity | 0.77 | 63.49 | 15.47 | 0.91 | −0.056 |
| lm | Prec. seasonality + field capacity | 0.77 | 63.67 | 15.65 | 0.91 | −0.084 |
| lm | MAP-PET | 0.72 | 64.36 | 16.34 | 1.00 | −0.150 |
| lm | MAP + Solar radiation | 0.76 | 64.55 | 16.53 | 0.93 | −0.095 |
| lm | Field capacity + nitrogen | 0.74 | 66.19 | 18.17 | 0.96 | 0.026 |
| lme | Field capacity + nitrogen | 0.78 | 68.15 | 20.13 | 0.74 | 0.026 |
| lm | MAP | 0.66 | 68.18 | 20.16 | 1.10 | −0.070 |
| lm | Field capacity | 0.61 | 71.10 | 23.08 | 1.18 | −0.011 |
| lme | Field capacity | 0.74 | 71.66 | 23.64 | 0.82 | −0.011 |
| lme | Prec. Seasonality + field capacity | 0.76 | 72.16 | 24.14 | 0.89 | −0.084 |
| lme | MAP | 0.85 | 74.82 | 26.80 | 0.54 | −0.070 |
| lme | MAP + field capacity | 0.86 | 75.16 | 27.14 | 0.57 | −0.056 |
| lme | Prec. seasonality | 0.75 | 76.15 | 28.13 | 0.80 | −0.116 |
| lme | MAP-PET | 0.83 | 76.88 | 28.86 | 0.61 | −0.150 |
| lme | MAP + prec. seasonality | 0.89 | 77.19 | 29.17 | 0.56 | −0.006 |
| lm | Nitrogen | 0.45 | 77.99 | 29.97 | 1.41 | −0.064 |
| lme | Field capacity + nitrogen + MAP | 0.85 | 78.15 | 30.13 | 0.59 | −0.052 |
| lme | MAP-PET + temp. during growing season | 0.89 | 78.41 | 30.39 | 0.49 | −0.051 |
| lm | Solar radiation + nitrogen | 0.45 | 81.14 | 33.12 | 1.41 | −0.064 |
| lme | MAP-PET + MAT | 0.56 | 81.90 | 33.88 | 0.55 | −0.087 |
| lm | Solar radiation | 0.29 | 83.10 | 35.08 | 1.60 | −0.022 |
| lme | Solar radiation | 0.84 | 86.49 | 38.47 | 0.69 | −0.022 |
| lme | Solar radiation + nitrogen | 0.84 | 89.59 | 41.57 | 0.69 | −0.064 |

*MAP* mean annual precipitation, *PET* potential evapotranspiration, MAP–PET water balance (mm), *MAT* mean annual temperature.
Field capacity and soil nitrogen content were log-transformed. Biome was used as random effect in linear mixed effects models. Each model was significant at $p < 0.01$.

**Table 2 Coefficient of determination ($R^2$) and root mean square error (RMSE) after excluding individual biomes from the ´best´ linear regression model.**

| Biome excluded | $n$ | $R^2$ | RMSE |
|---|---|---|---|
| Temperate conifer forest | 4 | 0.90 | 0.62 |
| Temperate broadleaf forest | 6 | 0.91 | 0.59 |
| (Sub-)Tropical moist broadleaf forest | 5 | 0.89 | 0.58 |
| (Sub-)Tropical savannas and woodlands | 3 | 0.82 | 0.64 |
| Boreal forest | 2 | 0.91 | 0.58 |

Each linear regression sub-model was significant at $p < 0.0001$. '$n$' refers to the number of sites in the respective biome that was excluded from the model.

determined by the degree of complementarity in crown architectures and the diversity and intermingling of tree sizes, with the latter depending on the shade tolerance of the species involved[26]. Complementarity in crown architectures and shade tolerance depend on the functional diversity and the range of plant functional types and strategies, which were shown to be greater in wetter rather than in drier environments (physiological tolerance hypothesis)[36]. In functionally diverse tree species communities, inherent differences in crown architecture between species may lead to greater complementarity and thus result in more complex forest structures (niche complementarity). Shade tolerance, for example, is inversely correlated with tolerance to other limiting factors, such as water limitation[37]. As such, shade-tolerant species are more frequently found in ecosystems where growth is not limited by factors other than light. Greater forest structural complexity in more humid climates can thus be partially attributed to the higher abundance of shade-tolerant trees of different sizes that cause vertical stratification and are able to co-exist and grow in light-limited under-story or mid-story canopy layers. Upper canopy layers are limited by maximum tree height and determine the available three-dimensional space that may be occupied. Tree height is constrained by water availability[38] (hydrological limitation hypothesis). Several studies have shown that canopy height increases with increasing annual precipitation (up to a certain threshold) and that water availability is a strong predictor of maximum canopy height[39]. Thus, relationships between water availability and forest structural complexity can be further attributed to mechanisms determining possible tree size. Against this background, the increase in forest structural complexity with increasing water availability most likely results from a combination of factors that are determined by water availability, for example functional diversity (species richness and complementarity in crown architectures), physiological tolerance to limiting factors (shade tolerance) and possible tree size (hydrological limitation). However, a more detailed understanding of relationships between functional diversity, specific functional traits in particular, and forest structural complexity is scarce. Recent advances in mapping functional diversity may allow for an improved understanding of linkages between functional diversity and forest structural complexity across spatial scales[40–42]. Identifying the functional drivers of forest structural complexity could help to further unravel the mechanistic underpinnings of

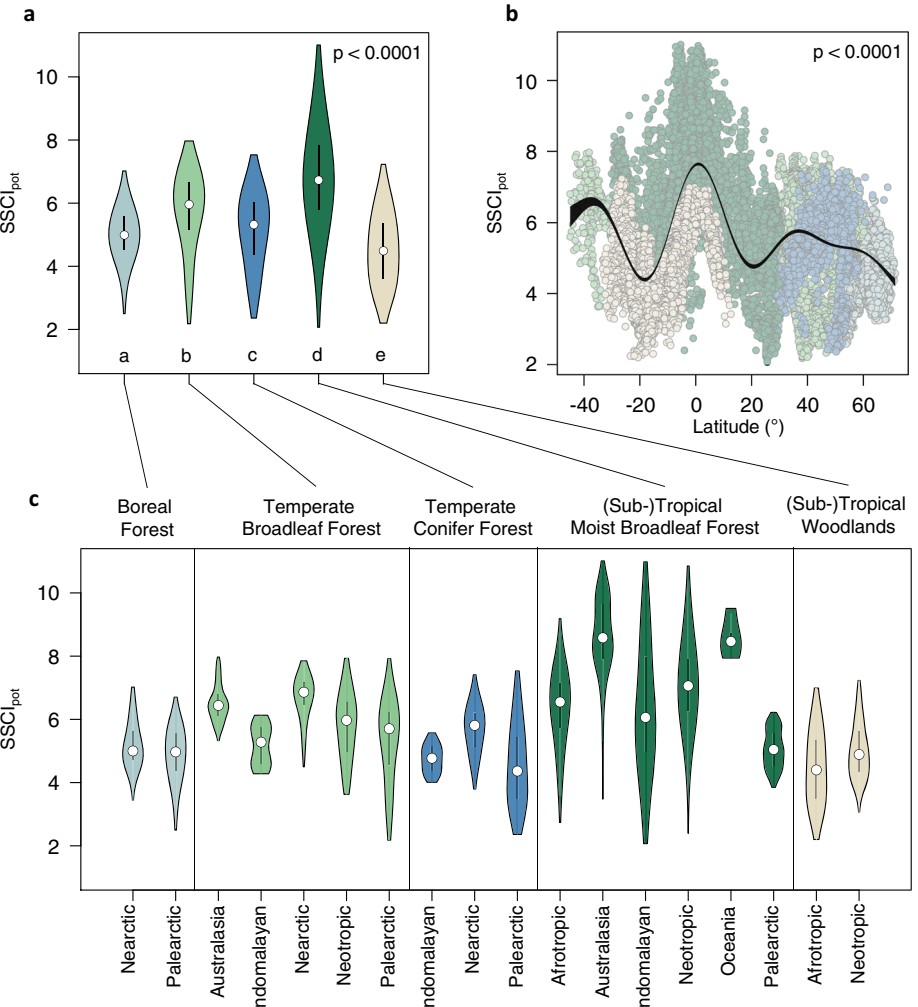

**Fig. 3 Global variability and latitudinal patterns of the potential structural complexity.** Globally modeled forest structural complexity ($SSCI_{pot}$), expressing the potential structural complexity across and within biomes (**a**), latitudes (**b**), and realms (**c**). Data points ($n = 21,851$) are samples based on a systematic global sampling grid with a distance of 50 km between points. Letters in **a** indicate significant differences in $SSCI_{pot}$ between biomes (one-way ANOVA, Tukey HSD post-hoc test, $p < 0.0001$). White dots mark the median, black lines the interquartile range and colored violins the probability density of the underlying distribution. The black band in **b** represents the 95% confidence interval of a thin-plate regression spline based on a generalized additive model ($p < 0.0001$).

relationships between climate and forest structural complexity in more detail.

Whether forest ecosystems develop complex three-dimensional structures further depends on the frequency, intensity and scale of disturbances. In order to first understand the climatic controls on forest structural complexity, we aimed at minimizing the modifying effects of disturbances on forest structure, by focusing our sampling on sites that represented primary forests of late-successional stages, thereby representing a climatic climax. However, natural disturbances are an integral component of forest ecosystem dynamics and play an important role in shaping forest structural complexity. Small-scale disturbances, like tree fall gaps, may promote structural complexity by creating favorable conditions for understory trees to develop[43]. Large-scale disturbances, such as fires or storms, modify forest structures by initially simplifying complex structures or suppressing its development[44]. In forests adapted to frequent fires, complex forest structures are typically spatially separated, resulting in a patchy distribution of single trees, tree clumps, and forest openings[45]. Thus, the spatial variability of forest structures may be partially

shaped by disturbances[46]. Mapping forest disturbance regimes worldwide, as it has already been done for Europe[47], could expand our work and enable the inclusion of disturbance regimes in modeling the dynamics of forest structural complexity. Furthermore, the variability in soil conditions can control the within-site spatial variability of forest structural complexity. The significant correlation between soil water holding capacity (field capacity) and forest structural complexity shown here underlined the fact that soil conditions may control small-scale variations of forest structural complexity. For example, forest structural complexity might deviate from our model predictions where low soil depth limits rooting space or where tree growth is negatively affected by permanent or temporal water-logging. The small-scale spatial variability of forest structural complexity is partially reflected in the variability of SSCI between plots at the respective study sites. The variability between plots may thus reflect differences in soil conditions and/or disturbance legacies. However, these small-scale differences in soil conditions could not be further addressed within the frame of this study, because the globally modeled soil data used here had a spatial resolution of 250 m.

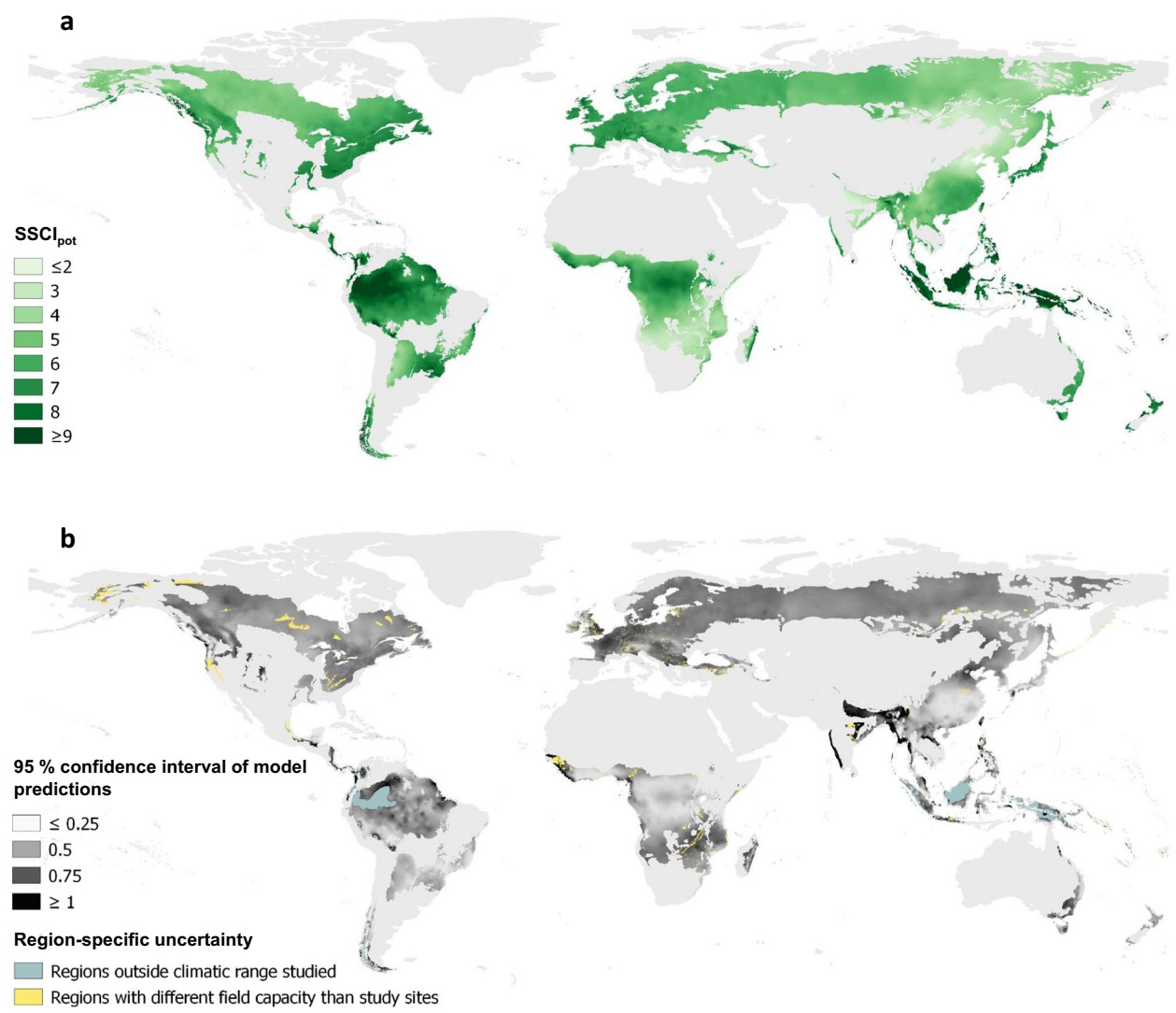

**Fig. 4 Global patterns of potential structural complexity. a** Potential structural complexity ($SSCI_{pot}$) in forest ecoregions across biomes. $SSCI_{pot}$ depiction was confined to biomes that were sampled within the frame of this study and are classified as forest or woodland ecoregion according to Olson et al. $SSCI_{pot}$ of Mediterranean Forests and Woodlands, Dry Broadleaf Forests, Tropical Conifer Forests and Mangroves is not shown here. Predictions are based on the WorldClim2 dataset for the years 1971–2000[33] and were made at 30 arcsecond resolution. **b** 95% confidence interval of $SSCI_{pot}$ model predictions. Regions outside the climatic range studied and regions with different soil conditions than our study sites are marked in light blue and yellow, respectively, because we cannot reliably quantify the uncertainty of model predictions for those areas.

Considering the potential mechanistic underpinnings of forest structural complexity, predicted changes of global precipitation and disturbance regimes strongly suggest changes in the global patterns of forest structural complexity in the future (see Supplementary Fig. 10). For example, a decrease in annual precipitation or an increase in seasonality may result in losses of functional and taxonomic diversity[10], especially where trees already operate close to their physiologically tolerable limits[48], or where a decrease in water availability reduces possible tree size or increases tree mortality[49]. In turn, these changes might feedback into changes in forest structural complexity. Several studies suggest that ongoing climate change is likely to result in either more frequent and/or more intense disturbance events such as wildfires, storms, droughts, and extreme temperature or precipitation events[3,50]. For example, in temperate central and western Europe, more frequent and intense dry periods during the growing season could result in a decrease in forest structural complexity, since the mortality of native tree species that are less adapted to prolonged dry periods might increase due to hydraulic failure[51]. In the tropics, impacts of altered precipitation regimes on the structural complexity of forests in the Amazon basin could be further amplified by altered vegetation-atmosphere feedbacks due to deforestation and increasing droughts[50]. In boreal forests, an expected increase in wildfire frequency and intensity might counteract positive effects of altered precipitation regimes and increasing temperatures on forest structural complexity[52]. Furthermore, different species and species communities within the same biome or ecoregion may respond differently to climate change, depending on their functional plasticity, adaptability and evolutionary history[53]. Thus, climate change-induced impacts on forest structure may differ between forests of different tree species compositions and diversity within the same biome even when changes in climatic conditions are similar[54]. Impacts of species-specific responses might be particularly severe where climate change results in higher mortality rates of specific species[55].

The structural complexity of undisturbed primary forests serves as important benchmark for forest management and forest restoration, as managing for complexity has been increasingly

recognized as an effective method to sustain a broad range of ecosystem functions and biodiversity in managed forest land-scapes[56]. Especially in the context of forest landscape restoration, structural complexity has become a recognized surrogate for restoration effectiveness[25,57]. The extent of primary forests is globally declining and remaining primary forests are increasingly threatened by anthropogenic disturbances[58,59].

Against this background, our global map of the potential structural complexity provides an urgently needed benchmark for ecologically oriented sustainable forest management and forest landscape restoration, including areas where primary forests were irretrievably lost[58,60]. The robustness of the best performing model used for mapping the potential structural complexity could be confirmed by a leave-one-out-cross-validation. Moreover, excluding entire biomes from the analysis, and thereby up to 30% of data points, did not substantially reduce its explanatory power. Still, the confidence in model predictions is constrained by an incomplete biogeographic, climatic and edaphic coverage of study sites and needs to be acknowledged as a limitation to model extrapolation (see Supplementary Fig. 1 for distribution of study sites and climatic range covered). Spatially explicit estimates of the potential structural complexity may be further improved by complementing the biogeographic and climatic coverage of study sites in further studies and by including disturbance regimes and small-scale variations in soil conditions in modeling approaches.

The potential structural complexity, quantified here as $SSCI_{pot}$, reflects the structural complexity of old-growth, primary forest, i.e., a climate-defined climax of forest structural complexity. It resembles the theoretical concept of 'potential natural vegetation', which describes the species composition a site would potentially have without anthropogenic disturbance[61]. It hence reflects the level of forest structural complexity that could potentially develop, regardless of whether the area is presently forested or has been deforested or degraded due to logging or land-use change. Consequently, we chose Olson et al.'s (2001)[35] map of forest and woodland ecoregions as our benchmark for mapping the potential structural complexity globally, since their map intends to "approximate the original extent of natural communities prior to major land-use change"[35]. The identified hotspots of high potential structural complexity coincided with hotspots of plant diversity[62] and differences in $SSCI_{pot}$ between biomes follow a similar pattern to differences in the species richness of vascular plants. For example, Borneo lowland rainforests, Choco-Darien moist forests and Fiji tropical moist forests feature the highest species richness of the evaluated ecoregions and also rank among the tropical and subtropical moist broadleaf forest ecoregions with the highest $SSCI_{pot}$ values in their respective realms. In boreal forests, Central Canadian Shield forests feature the highest species richness in the Nearctic realm and rank among the highest in $SSCI_{pot}$. However, whether structural complexity and plant diversity correlate on a global scale remains elusive. Better understanding these relationships requires to further take the actual structural complexity into account.

In contrast to the potential structural complexity, actual forest structural complexity is subject to the temporal and spatial dynamics of changes in species composition and anthropogenic and natural disturbances, which our map of potential structural complexity does not reflect. The currently ongoing Global Ecosystem Dynamics Investigation (GEDI)[32] provides satellite-borne LiDAR data of the earth's forest and may soon enable the mapping of the actual structural complexity of the world's forests[63]. Relating the actual to the potential structural complexity would help to better interpret the current state of forests worldwide, to improve the identification of intact forest landscapes of high conservation-value, to monitor the effectiveness of restoration efforts and to better understand impacts of forest

management or forest degradation on biodiversity and ecosystem functions.

Here we present evidence that the structural complexity of undisturbed primary forests is strongly correlated with annual precipitation and precipitation seasonality. Using detailed field measurements of forest structural complexity derived from terrestrial LiDAR and taking the structural complexity of primary forests as benchmark, we provide a global estimate of the potential structural complexity across biomes and realms. The resulting map can provide a reference for forest management and restoration, as well as to better determine the structural intactness of the world's forests. Our results also highlight the need to integrate forest structural complexity in modeling climate change impacts on biodiversity and ecosystem functions. Better predicting changes in biodiversity and ecosystem functions requires an in-depth understanding of the feedback mechanisms between changing climatic conditions, disturbance regimes, ecosystem resilience and forest structural complexity.

## Methods

**Study sites.** In total, we sampled 294 plots at 20 primary forest sites across five biomes, with two sites in boreal forests, six in temperate broadleaf forests, three in subtropical tree savannas and woodlands, four in temperate conifer forests and five in tropical moist broadleaf forests, following Olson et al.'s (2001)[35] classification of terrestrial biomes. Here, biomes are defined as "the world's major communities, classified according to the predominant vegetation and characterized by adaptations of organisms to that particular environment"[64]. In distinction, ecoregions are defined as "relatively large units of land containing a distinct assemblage of natural communities and species, with boundaries that approximate the original extent of natural communities prior to major land-use change"[35]. Detailed information on study site locations, ecoregions, soil types and number of plots is shown in Table 3. Study sites were selected to cover a broad climatic gradient across biomes and to represent dominant forest types within their respective biome (Supplementary Fig. 1). To avoid bias due to anthropogenic disturbance, we only selected sites that were considered primary forests according to the FAO definition of primary forests. Primary forests are defined as being naturally regenerated forests of native species showing no signs of human disturbances or activities[65]. The undisturbed state of selected sites could be either confirmed by scientific literature, local expert knowledge, or was highly likely due to exceptional remoteness and distance to human settlement.

**Field data collection and sampling design.** At each site, we systematically laid out sample plots of $100 \times 100$ m (1 hectare) in size with a distance of at least 200 m between plot centers. The number of plots varied between sites, depending on the variability of forest structure between plots, accessibility, and in some cases on the patch size of the undisturbed area, with an average of ~15 plots per site, and a total number of 279 plots. At each plot, we performed five systematically distributed single, terrestrial laser scans using a FARO Focus 120 or a FARO M70 (Faro Technologies Inc., Lake Mary, USA) to assess the surrounding 3D forest structure. The scanner was placed on a tripod in ~1.3 m above ground and set to scan a field of view of 300° degrees vertically and 360° horizontally with an angular step width of ~0.035°. The spatial information acquired during each scan was automatically stored in a 3D point cloud in the hardware specific format. Scan positions within each plot followed a "five on a dice"-approach, with one scan in the plot center, and four scans spaced 42 m from the plot center in the direction of the plot corners. This plot design has proven to be useful in several other studies conducted by the authors[4,24]. In addition, we used angle count sampling (also known as Bitterlich–Sampling) at each scan position to estimate stand basal area (m² ha⁻¹) using a dendrometer (see ref. [17] for details on angle count sampling)

**3D point cloud processing.** Each scan was imported to the hardware-specific software FARO SCENE (Faro Technologies Inc., Lake Mary, USA, v.7.1.1.81) and standard filter algorithms were applied to each scan file to erase stray and erroneous points from the point cloud. The filtered point cloud was then exported into a text file in.xyz format, storing the 3D information in a three-columned data frame with $x$-, $y$-, and $z$-coordinates. During scan export, the point cloud resolution was lowered to a sixteenth of the original resolution to allow for faster processing, better handling and to tailor the point cloud resolution needed for index computations. The reduction in resolution translates into an angular step width of ~0.14° between laser beams, if scans were made with a lower resolution in the first place. Still, a higher resolution during scan acquisition is needed to lower the percentage share of stray and erroneous points. The exported.xyz-files were then imported into Mathematica (Wolfram Research, Champaign, USA) to compute the stand structural complexity index (SSCI) after Ehbrecht et al.[11]. SSCI is based on the shape complexity of cross-sectional polygons derived from the 3D point clouds that is

**Table 3 Study sites.**

| Biome | Site | Ecoregion | n (plots) | Long. (°) | Lat. (°) | Elevation (m a. s.l.) | Soil type |
|---|---|---|---|---|---|---|---|
| Boreal forests | Fairbanks (USA) | Interior Alaska-Yukon lowland taiga | 25 | −148.11 | 64.80 | 198 | Gleysol |
| | Muddus (Sweden) | Scandinavian and Russian taiga | 15 | 20.06 | 67.05 | 532 | Podzol |
| Temperate broadleaf forests | Big Reed (USA) | New England-Acadian forests | 12 | −69.06 | 46.36 | 418 | Podzol |
| | Lagodechi Nature Reserve (Georgia) | Caucasus mixed forests | 12 | 46.32 | 41.85 | 798 | Cambisol |
| | Parque Tantauco (Chile) | Valdivian temperate forests | 9 | −73.79 | −43.02 | 134 | Andosol |
| | Reserva San Pablo de Tregua (Chile) | Valdivian temperate forests | 7 | −72.09 | −39.60 | 817 | Andosol |
| | Rožok (Slovakia) | Carparthian montane forests | 30 | 22.46 | 48.98 | 648 | Cambisol |
| | Uholka-Shyroki Luh (Ukraine) | Carparthian montane forests | 30 | 23.62 | 48.27 | 746 | Cambisol |
| Temperate conifer forests | Parque Nacional Villarica (Chile) | Valdivian temperate forests | 4 | −71.51 | −39.58 | 1269 | Andosol |
| | HJ Andrews Experimental Forest (USA) | Central-Southern Cascades forests | 10 | −122.22 | 44.23 | 543 | Cambisol |
| | Rockefeller Forest (USA) | Northern California coastal forests | 15 | −123.95 | 40.35 | 142 | Inceptisol |
| | Whitaker Forest (USA) | Sierra Nevada forests | 8 | −118.93 | 36.71 | 1641 | Alfisol |
| (Sub-)Tropical savannas and woodlands | Bwabwata National Park (Namibia) | Zambezian Baikiaea woodlands | 11 | 21.76 | −18.21 | 1035 | Arenosol |
| | Chobe Forest Reserve (Botswana) | Zambezian Baikiaea woodlands | 10 | 24.61 | −18.23 | 989 | Arenosol |
| | Khaudum National Park (Namibia) | Zambezian Baikiaea woodlands | 16 | 20.73 | −18.45 | 1082 | Arenosol |
| (Sub-)Tropical moist broadleaf forests | Budongo Forest Reserve (Uganda) | Albertine Rift montane forests | 10 | 31.53 | 1.73 | 1070 | Ferralsol |
| | Danum Valley (Malaysia) | Borneo lowland rain forests | 22 | 117.79 | 4.97 | 255 | Acrisol |
| | Maliau Basin (Malaysia) | Borneo lowland rain forests | 18 | 116.97 | 4.74 | 276 | Acrisol |
| | Saddle Road Forest Kipuka (USA) | Hawai'i tropical moist forests | 15 | −155.31 | 19.67 | 1433 | Andosol |
| | Colo-I-Suva Forest (Fiji) | Fiji tropical moist forests | 15 | 178.44 | −18.05 | 221 | Cambisol |

quantified using an estimate of the fractal dimension, based on a modified perimeter-to-area ratio, and is computed after McGarigal and Marks (1994) (see[11]). Cross-sectional polygons were constructed by splitting the point cloud into azimuthal sectors of ~0.14° and connecting the points with straight lines along the hemisphere, starting with the lowest point at 0°, to the zenith at 90°, and continuing from the zenith down to 180° in the opposite azimuthal sector. An angular step width of 0.14° then results in 1280 pairs of azimuthal sectors forming cross-sectional polygons, for which fractal dimension values of the 1280 cross-sectional polygons are then averaged to get a measure of structural complexity for each scan. However, the fractal dimension itself is a scale-independent measure of complexity and needs to be scaled in order to take stand size and vertical stratification into account. Mean fractal dimension values are scaled by using the natural logarithm of the effective number of layers (ENL), which quantifies the number of 1 m-thick vertical layers that are effectively occupied by foliage and woody components, as an exponent. ENL is computed by applying the inverse Simpson-Index to the vertical distribution of points, binned into layers of 1 m thickness (see ref. [19] for details). The SSCI used in this study has been successfully applied to quantify forest management impacts on forest structure[11], to quantify how forest structural complexity relates to tree species composition and mixture[24], to unravel relationships between structural complexity and the availability of tree microhabitats[66], to better understand effects of structural complexity on forest microclimate[67], and to quantify how restoration plantings with native tree species promote structural complexity and impact biodiversity in oil palm plantations[25]. Further explanations of how the index works, how it correlates with other metrics of forest structural complexity, as well as its limitations, are discussed in the Supplementary Information. The algorithm to compute SSCI using the statistical software environment R is available at https://github.com/ehbrechtetal/Stand-structural-complexity-index–SSCI[68]. Canopy height was derived from subtracting the lowest from the highest z-coordinate in the point cloud for each scan. Furthermore, we computed canopy openness by projecting the raw point cloud to a plane, using a stereographic projection. The percentage of canopy openness was then calculated for an opening angle of 60° from the laser scanner´s perspective.

**Climate and soil data**. The climate variables used to test relationships between climate and forest structure represent average values for the period 1971–2000 and were extracted from raster data provided in the WorldClim2 database, with a spatial resolution of 30 arcsecond (~1 km) (see ref. [33] for details). The respective variables were sampled for each plot using the coordinates of the plot center, and averaged across plots per site. Mean annual precipitation (MAP in mm), precipitation seasonality, expressed as coefficient of variation of monthly rainfall, mean annual temperature (MAT in °C) were used as provided by WorldClim2 database. Solar radiation (in kJ m$^2$ day$^{-1}$) and mean temperature during the growing season were computed as average values for the vegetation period. For boreal and temperate forests, the growing season was defined by the number of months with an average temperature of ≥5 °C. For tropical and subtropical forests and woodlands, growing season was defined by the number of months where precipitation was higher than half the potential evapotranspiration (PET)(sensu FAO, 1980). PET was derived from the raster images provided by the Global Aridity Index and Potential Evapotranspiration Database[69], which were also based on the WorldClim2 database[33]. To discuss impacts of climate change on forest structural complexity, we used projected climate data for 2070 as provided by the WorldClim2 database with a spatial resolution of 30 arseconds. The soil variables used to test relationships between edaphic factors and forest structure were derived from raster images of the SoilGrids database[70] (soil nitrogen content and cation exchange capacity) and soil profile data of the Regridded Harmonized World Soil Database v1.2[71] (field capacity). Soil nitrogen stocks and cation exchange capacity were sampled for each plot using the coordinates of the plot center and averaged across plots per site. Field capacity was derived from the soil profile that was closest to the study site center. Both databases provided soil variables for standard depth intervals (0−5 cm, 5−15 cm, 15−30 cm, 30−60 cm, and 60−100 cm). Relationships between soil variables and forest structure were then tested based on weighted means for 1 m soil depth.

**Statistical analyses**. All statistical computations were done using the open-source statistical software environment R, version 3.5.2 (R Development Core Team, 2019).

First, index values from single scans and basal area estimates based on angle count sampling were aggregated to plot means. Based on plot means, we estimated mean SSCI, canopy height, basal area, and canopy openness, and their standard errors for each site. We then used linear regression and linear mixed effect models to model effects of single climate variables on SSCI, canopy height, canopy openness or basal area. As a next step, we tested all possible combinations of single climate variables as predictors of forest structural metrics in linear and linear mixed effects models. To avoid collinearity in explanatory variables, we excluded variable combinations that showed an $r ≤ |0.7|$[34]. Linear mixed effect models were computed using the nlme R-package version 3.1–145. Since assumptions of normally distributed residuals and variance homogeneity were met and the data did not show any non-linear trends, we did not test other linear or non-linear, non-parametric models. We used a significance level of $p < 0.05$ for all analyses. Models were rejected, if one of the predictor variables was not significant. We then ranked

all models in increasing order according to their AIC$_c$ values (Table 1) and used the model with lowest AIC$_c$ value for further modeling. In addition, we used an automated model selection algorithm (MuMln R-package v.1.43.17) to cross-check whether manual and automated model selection yielded the same result for selecting the ´best´ model. To test for model robustness, we cross-validated the model using a leave-one-out-cross-validation approach based on the caret R-package (version 6.0-86). Furthermore, we excluded entire biomes from the analyses, thereby excluding up to 30% of data points, and evaluated model performance with the remaining biomes based on R² and RMSE (Table 2). In some cases, study sites were located less than 100 km apart from each other. Thus, we tested for spatial autocorrelation in model residuals using Moran´s I. Tests for spatial autocorrelation of model residuals were conducted using the spdep R-package version 1.1-3.

As the study sites covered a broad climatic range across, but not within biomes, we spanned a global sampling grid with 50 km distance between sample points across the sampled biomes (see Supplementary Fig. 8). The sampling grid was created using an R-code provided by Fehrmann et al.[72]. To avoid model extrapolation to areas that were not classified as forest or woodland within forest biomes, the resulting data frame was filtered to only contain sample points that fell within ecoregions that contained "forest", "woodland", "taiga", "chaco", "yungas", "várzea", or "campinarana" in their ecoregion name. We further filtered out ecoregions that additionally contained the terms "steppe", "tundra", "meadow", or "grassland", such as "East European Forest Steppe" or "Central US Forest-Grassland transition". For each of the resulting 22,614 sample points, we picked the MAP and precipitation seasonality values from the WorldClim2 dataset to predict the potential forest structural complexity (SSCI$_{pot}$) based on the model with lowest AIC$_c$. In order to control for outliers with very high MAP or precipitation seasonality in each biome, we plotted both variables against each other and estimated the contour of the 95% kernel density using the kernel density function in R. All points outside the 95% contour line, which is the polygon that contains 95% of the data, were excluded from further analyses (see Supplementary Fig. 9). This procedure was done for each biome separately. To visualize the climatic range of each biome and to check for the distribution of study sites within the global climatic range as shown in Supplementary Fig. 1, we clipped convex hulls to the points remaining after 95% kernel density estimation. The remaining 21,581 sample points were then used for further analyses. Differences between biomes were tested using a one-way ANOVA and a subsequent Tukey HSD test. The latitudinal pattern in forest structural complexity was evaluated using generalized additive modeling (mgcv R-package, version 1.8-31).

To map the potential forest structural complexity SSCI$_{pot}$ on a global scale, we created a raster image with the same resolution as the raster image provided by WorldClim2 (30 arcsecond resolution) using the geostatistical software SAGA (version 7.4.0), and predicted SSCI$_{pot}$ for each pixel in the image that fell within the ecoregions described above. To avoid model extrapolation, all pixels for which an SSCI$_{pot}$ was predicted to be higher or lower than the highest or lowest SSCI of our study sites, were classified conservatively as SSCI$_{pot}$ ≥ 9 or ≤ 2.

To test whether hotspots of high structural complexity coincided with hotspots of biodiversity, we combined the dataset provided by ref. [62] with our model-based estimates of potential structural complexity (SSCI$_{pot}$) for each forest ecoregion.

**Reporting summary**. Further information on research design is available in the Nature Research Reporting Summary linked to this article.

## Data availability
The data supporting the findings of this study are available at https://doi.org/10.25625/HYIMZG. Maps of potential structural complexity and confidence intervals are available at https://doi.org/10.25625/9NPEQA. The climate data used in this study is publically available at worldclim.org[33], soil data used in this study is available at soilgrids.org[70] and the Regridded Harmonized World Soil Database[71].

## Code availability
The code for computing the forest structural complexity index (SSCI) used in this study is publicly available at https://github.com/ehbrechtetal/Stand-structural-complexity-index–SSCI (https://doi.org/10.5281/zenodo.4295910).

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

## Acknowledgements

For financial support, we thank the German Research Foundation (DFG) (SE 2383/1-1; SE2383/4-1; SE2383/5-1), the Eva-Mayr-Stihl foundation (Project No. 180124), and the Dr. Erich Ritter-Foundation (T0021/29427/2017). For granting research permits and access to research sites, we thank the Sabah Biodiversity Centre, Sabah Biodiversity Council, Danum Valley and Maliau Basin Management Committee (Access License Ref. No. JKM/MBS.1000-2/2 JLD.9 (45), the Carpathian Biosphere Reserve, the Environmental Protection Office in Prešov, the administration of Poloniny National Park, the Agency of Protected Areas and the Lagodekhi Protected Areas administration and Budongo Conservation Field Station. For logistical, administrative and/or fieldwork support, we thank Mr. Geoffrey Muhanguzi, David Eryenyu, John Paul Okimat, Moses Businge, Douglas Ryan, Vasyl Lavnyy, Yuriy Berkela, Myroslav Kabal, Peter Jaloviar, Stanislav Kucbel, and Mrs. Natia Shalvashvili. The support from the South-East Asian Rainforest Research Partnership is gratefully acknowledged (SEARRP Project No. 18017). Martin Ehbrecht thanks Prof. Dr. Majeliiwa for the invitation to conduct parts of the fieldwork in Budongo Forest Reserve. Daniel Soto is thankful for the support of Fondecyt 11181140. We conducted parts of the research at HJ Andrews Experimental Forest, which is funded by the US Forest Service, Pacific Northwest Research Station.

## Author contributions

M.E., C.A., D.S., and P.A. designed the study, M.E., P.A., D.S., K.W., and M.S. coordinated and implemented data collection, D.S. processed the terrestrial laser scanning data, M.K. computed the world maps and supported geo-spatial data analyses, M.E. analyzed the data and wrote the first draft of the manuscript, with all authors contributing substantially to revisions.

## Funding

## Competing interests

The authors declare no competing interests.
