## [Peer Review File · Nature Communications]

Reviewer Comments, first round -

Reviewer #1 (Remarks to the Author):

Overall, this manuscript shows that ground-based lidar measurements of forest structural complexity (FSC) correlate strongly at the site scale with mean annual precipitation and precip seasonality around most of the world. I am impressed by the amount of work that went into this project. The writing of the paper is strong and the analyses are clear.

Nevertheless, I am struggling to get excited about this paper as it is currently presented, and I'll try to articulate why I think that is here. First, FSC is a very conceptually complex thing - at the local scale it reflects potential biomass (therefore climate), disturbance history and forest age, biogeography (species distributions), geologic history, and more. As such, it is unlikely for there to be one driver of FSC, and FSC is often used as a symptom of these other processes, not mechanistically connected to only one thing. So, given that, it is remarkable that this manuscript finds that FSC is so strongly correlated with MAP and precip seasonality. But the manuscript never reaches so far as to really address why that might be. What are the potential mechanisms that would lead FSC to be so strongly connected to precip patterns? One recommendation related to these comments would be to think about moving much of the text from paragraphs 2 and 3 from the discussion up to the introduction, and to generally focus the introduction more on what FSC really driven by and less on what its been shown to correlate with.

I am also surprised by this manuscript's result that climate was not correlated with canopy height or basal area, which are well established patterns (discussion of this is currently mostly buried in the supplement). Why might this be the case? Similarly, given the complexity of SSCI, and the wide range of other FSC metrics out there, how does SSCI compare? Is it possible that these results are SSCI specific?

Finally, I am not sure about the utility of the SSCI in 2070 approach, or I think it needs to be handled more carefully. To suggest that SSCI is going to dramatically change in 50 years based only on CMIP predictions of changes in precip (which are HIGHLY uncertain) ignores the important mechanistic underpinnings of FSC, which develops over decades or even centuries. This type of analysis strikes me as being similar to early climate envelope modeling of species distributions which would indicate that tropical species would be growing near the poles in a few decades, ignoring all of the other ecological processes that determine where species grow. I get that this is 'potential' SSCI, but without a clear mechanistic explanation for why MAP and prec. seasonality explain so much variation in SSCI, I am skeptical that this is a useful approach.

Specific comments:

line 40: "we find that *of the variables tested* the global..." - or something. I think it is noteworthy earlier in the manuscript that this study only tested climate drivers.

Intro paragraph 1 (and throughout): This intro leaves me asking 'so what'? and never really articulates why FSC is interesting or important from a mechanistic perspective.

line 110: 'We hypothesized...' why?

line 140: just to check, this should be $r < |0.7|$ right? a correlation of -0.8 is strong.

line 147: Moran's I - only for best model? this could be added as a column for all the models in table 1.

line 160: significant *linear* relationships?

line 164: I would avoid using R syntax in the main text of a paper - not everyone knows what this means.

line 171 (or in methods): should clarify or define use of 'ecoregion' vs 'biome'

line 256: see lots of literature on California plants, both remotely sensed and in-situ

line 284: hotspots of *plant* biodiversity. Also, it would be much more compelling if this were quantified, which it seems like it would be easy to do. As written it seems like just a selection of 'cherry picked' examples.

line 302: why weren't edaphic factors considered? or forest age? there are lots of global data sets now that would have been interesting to include.

line 347: 'driven by' -> 'correlated with'

line 357: LOTS of work focuses on relationships between plant functional traits and climate. a more specific suggestion would be more compelling.

line 364: biomes *and ecoregions* - would be useful to define there here.

line 448: would it be more clear to say 'was higher than half the PET'? if that is what this means?

line 474: again, I would avoid R syntax

line 484: I think this paragraph would benefit from a "To ..." sentence at the beginning explaining why this gridding needed to be done.

Overall, I think many of the supplemental figures aren't referenced in the main text, and there is quite a bit of information in the supplement that deserves more attention (or at least referred to in the text).

Reviewer #2 (Remarks to the Author):

The authors characterized forest structural complexity using terrestrial laser scanning in boreal, temperate, subtropical and tropical forests, and found that structural complexity is varying with precipitation and precipitation seasonality. The article is very well written and covers an important, relevant and timely topic of how structural diversity is distributed across biomes and what could possibly drive it. However, I have three major concerns about the conclusions made from the measurements they used: first on the prediction of structural complexity change under climate change based on a simple correlation at one point in time, second on the transferability of the results from a few plots to a global map (with major underrepresentation of certain regions, like a complete lack of sampling in the Amazon), and third on the TLS measurements in dense forests based on very few scans, where usually tens or hundreds of co-registered scans are needed to fully characterize the 3D structure without occlusion.

First, I think it is tricky to imply a causal relationship from the correlation the authors found. What if structural complexity is driven by other factors that happen to co-vary across precipitation gradients of the sites explored? How can you be sure that a change in precipitation will lead to a change in structural complexity, which is not driven by external factors?

Additionally, can you explain in more detail in the introduction how you define structural complexity and what are the main structural components of it (e.g. height, gap distribution, etc)?

The high-level explanation of structural complexity and its relevance is great, but doesn't reveal how the TLS measurements are actually used to derive it. There are many ways to characterize structural diversity, and they may influence the meaning of it (is it mostly horizontal or vertical variability, mostly height variability or gap distribution within the canopy that matters). See for example LaRue et al 2019 for different definitions of structural diversity:

LaRue, E. A., Hardiman, B. S., Elliott, J. M., & Fei, S. (2019). Structural diversity as a predictor of ecosystem function. *Environmental Research Letters*, 14(11), 114011.

<https://doi.org/10.1088/1748-9326/ab49bb>

On the other hand, the description of the SSCI in the Methods is very technical, and it's hard to imagine what the fractal dimension means and how it is related to more traditional measures such as the (horizontal and vertical) variability of functional traits, or variability of height or gap distribution. Also, if water availability is the major driver of structural complexity, then it would be crucial to include soil variables in the analyses. There are some global soil data sets, like SoilGrids, that could be used for that.

Second, I think it is problematic to extrapolate the findings to a global map. The authors provide leave-one-out-cross-validation. However, this kind of "validation" is only as good as the sampling used. If there are no plots for example in the Amazon or in central Europe, then there is no way to judge the uncertainties or quality of the extrapolation to those areas. The forest structure is quite different between Borneo and the Amazon, with huge variability within the regions. For example, I would expect a higher structural complexity in the eastern Amazon than in the west. Also, the tallest trees in the Amazon are found in the north-east (>70 m), which should indicate high structural complexity with multi-layered canopies and emergent trees 60-90 m tall, see: Gorgens, E., Nunes, M. H., Jackson, T., Coomes, D., Keller, M., Reis, C. R., ... Ometto, J. P. (2020). Resource availability and availability and disturbance shape maximum tree height across the Amazon University of Helsinki. *BioRxiv*. <https://doi.org/10.1101/2020.05.15.097683>
Gorgens et al. 2020 found that wind speed and soil clay content play important roles besides precipitation and radiation. I would assume that wind and soil (eg the combination of shallow soils and strong winds) are also important drivers of forest structure and structural complexity, additionally to the factors the authors considered. The relationship between structural complexity and height is reported to be hump-shaped (Figure SI2), but this relationship seems to be strongly driven by the Rockefeller forest, whereas tropical forest canopies with tree heights over 60 m seem to be lacking. The Rockefeller forest is a coastal redwood forest with some of the tallest trees on Earth, which is not necessarily the most representative site for temperate conifer forests (for example assuming that those findings would be transferable to temperate conifer forests in Europe or Russia). Therefore, I think an extrapolation to a global map without taking into account additional potential explanatory variables and without having any samples in the Amazon, Asia, Russia or Australia does not seem reasonable. The discussion is well written and addresses some of the issues. I think the study would still be highly relevant and more plausible, if the findings are reported just for the regions that have been sampled.

Third, how do the authors validate structural complexity and the influence of occlusion using their TLS measurement approach? Wilkes recommend about 20 m distance between measurements to optimize the scan patterns, and Schneider et al. 2019 found that even with a very dense scan pattern of about 10-15 m and 89 scan locations to cover 1 ha of Bornean rainforest, there still remains about 30-90% occlusion in the upper canopy:

Wilkes, P., Lau, A., Disney, M., Calders, K., Burt, A., Gonzalez de Tanago, J., ... Herold, M. (2017). Data acquisition considerations for Terrestrial Laser Scanning of forest plots. *Remote Sensing of Environment*, 196, 140–153. <https://doi.org/10.1016/j.rse.2017.04.030>

Schneider, F. D., Kükenbrink, D., Schaeppman, M. E., Schimel, D. S., & Morsdorf, F. (2019). Quantifying 3D structure and occlusion in dense tropical and temperate forests using close-range LiDAR. *Agricultural and Forest Meteorology*, 268(December 2018), 249–257. <https://doi.org/10.1016/j.agrformet.2019.01.033>

The attached figure shows how occlusion increases with the reduction of scan positions within a 1 ha forest plot in Borneo, with about 40% occlusion when only 6 scan positions are used.

With only 5 positions spaced 40 m apart, I wonder how the authors are capable of characterizing tropical forest structural complexity. It seems to me that there might be a systematic under-sampling of the upper canopy, with potentially large parts that remain unexplored/occluded. How does the method perform in tropical forests or in a Redwood forest, where the sampling of the upper canopy is very challenging? Do the authors have a way to quantify/report occlusion or a way to validate the structural complexity? Also, it seems that the calculation of the SSCI is even based on single scans, which is very problematic in tropical rainforests. Usually, only by adding many different observation positions together, one can get a clearer picture of the overall structure of the forest. A single scan has very limited reach in dense tropical forests (and probably even less with a phase shift vs. a time of flight scanner), and would mostly tell something about the structure of the understory and midstory.

In conclusion, I would recommend to focus on the conclusions that can be directly drawn from the sites measured, for example moving supplementary figure 1 to the main text to show the distribution of the plots and representativeness of climate regions, and move the global maps to the supplementary as "experimental results", unless a validation can be provided for the regions that were not sampled. I would also expect some discussion of the measurement approach, including potential limitations, and possible demonstration of the applicability of the approach in tropical rainforests.

Two detailed comments:

L255-258: But see Harrison et al. 2020 for a recent reference:
Harrison, S., Spasojevic, M. J., & Li, D. (2020). Climate and plant community diversity in space and time. *Proceedings of the National Academy of Sciences of the United States of America*.
<https://doi.org/10.1073/pnas.1921724117>

L266: But there might also be complex feedbacks, for example frequent natural fire disturbance can lead to increased local forest structure variability that make forests more resilient against tree mortality by fires, see:

Koontz, M. J., North, M. P., Werner, C. M., Fick, S. E., & Latimer, A. M. (2020). Local forest structure variability increases resilience to wildfire in dry western U.S. coniferous forests. *Ecology Letters*, *ele.13447*. <https://doi.org/10.1111/ele.13447>

Reviewer #1 (Remarks to the Author):

We thank reviewer 1 for the critical view on the manuscript and the suggestions for improvement. The most important changes related to comments by reviewer 1 are:

- Addressing the potential mechanistic underpinnings of forest structural complexity in the introduction (see revised Figure 1) and discussing potential explanations for our findings (in relation to the potentially underlying mechanisms) in the discussion (Line 56- 62, 103 – 106, 262 – 289).
- We included soil variables in a comprehensive re-analysis of the dataset. Also, we have added a further site from a further realm, for which the data was not available during original submission. Thereby we could confirm the robustness of our findings (Line 128 – 130, 145 - 153, 170).
- We have revised parts of the discussion to be more precise about the utility of our predictions how climate change will affect the potential structural complexity to avoid misinterpretations (Line 362 -406).

Overall, this manuscript shows that ground-based lidar measurements of forest structural complexity (FSC) correlate strongly at the site scale with mean annual precipitation and precip seasonality around most of the world. I am impressed by the amount of work that went into this project. The writing of the paper is strong and the analyses are clear.

Nevertheless, I am struggling to get excited about this paper as it is currently presented, and I'll try to articulate why I think that is here. First, FSC is a very conceptually complex thing - at the local scale it reflects potential biomass (therefore climate), disturbance history and forest age, biogeography (species distributions), geologic history, and more.

- We fully agree that FSC is affected by a multitude of different factors including disturbance history, forest age (or successional stage), biogeography and geologic history.
- Hence, we focused our study on old-growth, primary forests only, i.e. forests that represent late-successional, climax forest structures, thereby aiming at minimizing the effect of confounding factors. This is also the reason why we speak of 'potential' structural complexity, which is meant to describe a climatic climax of FSC (in analogy to the concept of potential natural vegetation). To further address the reviewer's comment, we have now included a revised version of figure 1 that outlines the abiotic and biotic determinants of forest structural complexity (Line 103 – 106).
- We think that forest age is an ambiguous measure, since old-growth, primary forests are (in most cases) uneven-aged. Whether the oldest trees represent forest age is at least doubtful, as long as the disturbance history is not known. And for most of the primary forests around the world, our knowledge of disturbance histories and legacies is largely incomplete. In that sense, our study aims to first understand the climatic (and now edaphic) controls of FSC. Based on that, understanding how disturbances, stage of pedogenesis or successional stage shape or modify FSC is seen as the next step.
- In our study, we aimed at addressing the effects of biogeography through our approach to sample different major forest types (with different species compositions) within each studied biome.
- Geologic history most likely affects FSC due to differences in pedogenesis. To address this, we have now included soil data in the revised version (Line 128 – 130, 145 – 153).
- We disagree that FSC reflects potential biomass. Structurally complex forests such as tropical rainforests usually have high biomass stocks, but forests with low biomass stocks are not

necessarily low in structural complexity. Vice versa, forests with high biomass stocks, such as Coastal Redwood forests, show a rather low structural complexity. Based on our data, we cannot confirm that structural complexity and biomass correlate.

As such, it is unlikely for there to be one driver of FSC, and FSC is often used as a symptom of these other processes, not mechanistically connected to only one thing. So, given that, it is remarkable that this manuscript finds that FSC is so strongly correlated with MAP and precip seasonality. But the manuscript never reaches so far as to really address why that might be. What are the potential mechanisms that would lead FSC to be so strongly connected to precip patterns? One recommendation related to these comments would be to think about moving much of the text from paragraphs 2 and 3 from the discussion up to the introduction, and to generally focus the introduction more on what FSC really driven by and less on what its been shown to correlate with.

- Thanks for the recommendation. We have now expanded the introduction on the potential drivers of FSC from a mechanistic perspective. To address this concern, we have revised the manuscript as follows:
 - We revised figure 1, which is now depicting the abiotic and biotic determinants of structural complexity (Line 128 – 130).
 - We revised the introduction, which is now introducing in more detail the determinants of forest structural complexity and why it could be related to climate (Line 56 – 62, 93 – 98).
 - We revised the discussion by adding an additional paragraph, in which potential mechanisms why forest structural complexity is so strongly related to climate are being discussed (Line 262 – 289).
- However, with our dataset, we are not able to test hypotheses with regard to specific mechanisms. We are not aware of any other study that has looked at relationships between climate and structural complexity so far. Thus, we understand our study rather as a first important step towards establishing a baseline and developing a more mechanistic theoretical framework.
- We agree that it is unlikely for there to be one single driver of FSC. As mentioned in the discussion, we assume that FSC emerges from interactions between different factors that are determined by precipitation and seasonality. Still, annual precipitation and precipitation seasonality explain ~90% of variance in our dataset. In the revised version, we now try to focus more on why that might be the case.

I am also surprised by this manuscript's result that climate was not correlated with canopy height or basal area, which are well established patterns (discussion of this is currently mostly buried in the supplement). Why might this be the case?

- This is probably due to the fact that we have Coastal Redwood forests in our dataset that are characterized by an immense canopy height and basal area, but that grow on sites – in a global comparison - with rather medium levels of annual precipitation (1200 – 1800 mm). When excluding the temperate conifer forest biome from the analysis, correlations between canopy height and precipitation become significant. However, please note that excluding this biome (or the Coastal Redwood sites) from the FSC analysis does not significantly affect the model outcomes with regard to structural complexity (see revised Table 2 in manuscript).
- Since relationships between climate and canopy height or basal area are intensively studied and well-known relationships and because they are not in the focus of this study, we opted to mention them in the SI.

Similarly, given the complexity of SSCI, and the wide range of other FSC metrics out there, how does SSCI compare? Is it possible that these results are SSCI specific?

- SSCI correlates well with other measures of FSC, such as Zenner & Hibbs (2000) Structural Complexity Index, which is up to today the most frequently cited index of FSC, as well as with the Gini-Coefficient of tree diameters (see Figure 1 here, which is also included in the *Supplementary Information* (Figure SI.2)), which is often used to characterize structural diversity or complexity (see Lexerod & Eid 2006). SSCI also correlates well with the tree size differentiation index (see Pretzsch, 2009, for further details).

Figure 1. Relationships between SSCI and other metrics of forest structural complexity. (A) Stand structural complexity index (Zenner & Hibbs, 2000), (B) Gini-Coefficient of tree diameters (see Lexerød & Eid 2006), and (C) tree size differentiation index (see Pretzsch 2009). Data points are based on 91, fully inventoried, stem-mapped forest plots of 1 hectare. Regressions are significant at $p < 0.0001$. Forest plots represent differently managed temperate broadleaf and conifer forests of Central Europe (shelterwood system, selectively logged and unmanaged, old-growth). SSCI values represent mean values of five systematically distributed single scans per plot (same sampling design as in the study under review). The shown correlations between SSCI and other measures of structural complexity are stronger than in the original publication, because they are based on fully inventoried and completely stem-mapped plots (in contrary to the results shown in the original publication, where trees smaller than 7 cm DBH were not accounted for). Data of tree attribute-based metrics of forest structural complexity is published in Schall et al. (2018).

- We cannot judge whether the results are SSCI specific, because we do not have other measures available for this study. However, we draw our confidence in SSCI from the above mentioned relationships with other FSC metrics and the numerous studies that have used SSCI so far.

Finally, I am not sure about the utility of the SSCI in 2070 approach, or I think it needs to be handled more carefully. To suggest that SSCI is going to dramatically change in 50 years based only on CMIP predictions of changes in precip (which are HIGHLY uncertain) ignores the important mechanistic underpinnings of FSC, which develops over decades or even centuries. This type of analysis strikes me as being similar to early climate envelope modeling of species distributions which would indicate that tropical species would be growing near the poles in a few decades, ignoring all of the other ecological processes that determine where species grow. I get that this is 'potential' SSCI, but without a clear mechanistic explanation for why MAP and prec. seasonality explain so much variation in SSCI, I am skeptical that this is a useful approach.

- As the reviewer has pointed out, we do not suggest that SSCI is going to change dramatically by 2070, but rather the potential for certain levels of forest structural complexity. We cannot model how the 'actual' structural complexity will change without taking the mechanistic underpinnings of FSC into account.
- The change map basically depicts how mean annual precipitation and precipitation seasonality change according to the currently available CMIP predictions and how these translate into the climatically-defined potential for certain levels of forest structural complexity.
- We think that this approach is useful, because larger differences between current and future 'potential' structural complexity imply a higher probability of changes in ecosystem structure and functioning. Spatially explicit knowledge about changes in the potential for certain levels of forest structural complexity is crucial for better adapting forest management and restoration efforts to climate change, since current management and restoration practices are those that shape the structure of future forests.
- We agree that this approach needs to be handled carefully. To avoid misinterpretations, we now try to highlight and discuss the limitations in more detail (Line 238 – 243, 372 – 406).

Specific comments:

line 40: "we find that *of the variables tested* the global..." - or something. I think it is noteworthy earlier in the manuscript that this study only tested climate drivers.

- We are not exactly sure what is meant here. Line 40 is part of the abstract and there is not really an opportunity to mention earlier which variables have been tested.

Intro paragraph 1 (and throughout): This intro leaves me asking 'so what?' and never really articulates why FSC is interesting or important from a mechanistic perspective.

- We have re-written the introductory paragraph of the introduction by trying to articulate more why FSC is interesting from a mechanistic perspective (Line 56-62).

line 110: 'We hypothesized...' why?

- We now try to outline more 'why', by introducing the determinants in more detail. Against this background, the reason why we hypothesize what we hypothesize should be more apparent now.

line 140: just to check, this should be $r < |0.7|$ right? a correlation of -0.8 is strong.

- Thanks for the hint. Corrected.

line 147: Moran's I - only for best model? this could be added as a column for all the models in table 1.

- In the revised version, Moran's I is now shown for each model.

line 160: significant *linear* relationships?

- We did not find significant relationships, neither linear nor non-linear.

line 164: I would avoid using R syntax in the main text of a paper - not everyone knows what this means.

- Agreed. However, we have tried to avoid mathematical formulas and to keep it simple by simply mentioning the explanatory variables and type of model used.

line 171 (or in methods): should clarify or define use of 'ecoregion' vs 'biome'

- We have adapted the terminology of ecoregion and biome from Olson et al. (2001), which we have also used for classification. To follow the reviewer's suggestion, we now briefly define what is meant in the methods section (Line 431 – 436).

line 256: see lots of literature on California plants, both remotely sensed and in-situ

- Thanks. We were not aware of that literature and now cite Harrison et al. (2020).

line 284: hotspots of *plant* biodiversity. Also, it would be much more compelling if this were quantified, which it seems like it would be easy to do. As written it seems like just a selection of 'cherry picked' examples.

- In fact, this statement is based on data. Kier et al. (2005) provided their estimates of vascular plant diversity in the supplement of their paper. We have combined their dataset with ours. It is only 'cherry picking' in the sense that the named examples represent forest ecoregions of larger extent (which is why we write "...rank among the highest..")
- We now mention in the methods section how the hotspots were determined (Line 577 – 579).

line 302: why weren't edaphic factors considered? or forest age? there are lots of global data sets now that would have been interesting to include.

- We now consider soil variables in our analyses (soil water holding capacity (field capacity), soil nitrogen, cation exchange capacity).
- We doubt that forest age is an appropriate measure here, as most old-growth, primary forests are uneven-aged. As mentioned earlier, the oldest trees in old-growth forests do not necessarily represent forest age. We would agree if the study was about managed forests or forests of different successional stages.

line 347: 'driven by' -> 'correlated with'

- Changed accordingly.

line 357: LOTS of work focuses on relationships between plant functional traits and climate. a more specific suggestion would be more compelling.

- We have revised the respective paragraph and now take further literature on that topic into account.

line 364: biomes *and ecoregions* - would be useful to define there here.

- See response to comment on line 171

line 448: would it be more clear to say 'was higher than half the PET'? if that is what this means?

- Agreed. Changed accordingly.

line 474: again, I would avoid R syntax

- Agreed. Changed accordingly.

line 484: I think this paragraph would benefit from a "To ..." sentence at the beginning explaining why this gridding needed to be done.

- Agreed. Changed accordingly.

Overall, I think many of the supplemental figures aren't referenced in the main text, and there is quite a bit of information in the supplement that deserves more attention (or at least referred to in the text).

- Figure SI.3, SI.4 and SI.5 (now Figure SI.4, SI.5 and SI.6, respectively) were not referenced so far. We have now referenced these figures where we found it appropriate.

Reviewer #2 (Remarks to the Author):

We thank reviewer 2 for the very thorough review of our manuscript and the many critical points that were highlighted. We aimed at addressing each point very thoroughly. The most important changes to the manuscript related to comments by reviewer 2 are:

- We included soil variables in a comprehensive re-analysis of the dataset, which did not improve our models and thereby confirmed the robustness of our findings (Line 128 – 130, 145 - 153, 170).
- A further site from a further realm was added, for which the data was not available during original submission (Tropical moist broadleaf forest in Fiji Islands, Oceania). Including an additional site did not affect the model outcomes and further underlined the robustness of our findings.
- We have added a further map depicting the uncertainty in model predictions and outlined regions for which the uncertainty cannot be quantified reliably, thereby aiming at acknowledging the limitations of the study (Line 234 – 237).
- A further chapter was added to the *Supplementary Information*, providing detailed information on index construction, its correlation with other structural complexity measures, and a discussion of its limitations (Line 18 – 68 in *Supplementary Information*).

The authors characterized forest structural complexity using terrestrial laser scanning in boreal, temperate, subtropical and tropical forests, and found that structural complexity is varying with precipitation and precipitation seasonality. The article is very well written and covers an important, relevant and timely topic of how structural diversity is distributed across biomes and what could possibly drive it. However, I have three major concerns about the conclusions made from the measurements they used: first on the prediction of structural complexity change under climate change based on a simple correlation at one point in time, second on the transferability of the results from a few plots to a global map (with major underrepresentation of certain regions, like a complete lack of sampling in the Amazon), and third on the TLS measurements in dense forests based on very few scans, where usually tens or hundreds of co-registered scans are needed to fully characterize the 3D structure without occlusion.

- We agree that the incomplete biogeographic coverage of our study sites is a limitation of our study. Due to logistic constraints, we were not able to cover further sites.
- In the revised version, we have aimed at explicitly highlighting and acknowledging the limitations of model extrapolation in more detail (Line 234 – 237, 291 – 316, 327 - 338, 372 – 408). Despite the limitations, we think that our model is robust enough to allow for an extrapolation. We try to outline why in the response to the reviewer’s specific comments on that topic (page 11 in this document) and in Line 327 -338 in the revised manuscript.
- Furthermore, we now try to be more clear about the interpretation of climate change-induced changes in the potential structural complexity (Line 238 – 243).
- In the revised version, we have complemented the global maps by a map depicting the uncertainty in model predictions (95 % confidence interval), including regions for which we cannot reliably quantify the uncertainty due different climatic or edaphic conditions than our study sites (Line 234 – 237).
- Moreover, we now provide a detailed explanation why we think that our TLS-measurement approach works also in dense tropical forest in the following (page 13 – 15 in this document). Parts of our explanation are now included in the revised version of the manuscript (*Supplementary Information*, Line 18 – 68)

First, I think it is tricky to imply a causal relationship from the correlation the authors found. What if structural complexity is driven by other factors that happen to co-vary across precipitation gradients of the sites explored? How can you be sure that a change in precipitation will lead to a change in structural complexity, which is not driven by external factors?

- Indeed, it is problematic to imply causal relationships from the correlations we have found, which we have actually tried to avoid. In the revised version, we now try to use more cautious wording and now only speak of “correlated with” (or similar) throughout the manuscript.

Additionally, can you explain in more detail in the introduction how you define structural complexity and what are the main structural components of it (e.g. height, gap distribution, etc)? The high-level explanation of structural complexity and its relevance is great, but doesn’t reveal how the TLS measurements are actually used to derive it. There are many ways to characterize structural diversity, and they may influence the meaning of it (is it mostly horizontal or vertical variability, mostly height variability or gap distribution within the canopy that matters). See for example LaRue et al 2019 for different definitions of structural diversity:

LaRue, E. A., Hardiman, B. S., Elliott, J. M., & Fei, S. (2019). Structural diversity as a predictor of ecosystem function. *Environmental Research Letters*, 14(11), 114011. <https://doi.org/10.1088/1748-9326/ab49bb>

- We define forest structural complexity by the heterogeneity of biomass or foliage distribution in three-dimensional space (sensu Gough et al. (2020)). A high heterogeneity of biomass or foliage distribution in three-dimensional space is realized through a high diversity and intermingling of different tree sizes and crown architectures, resulting in a greater connectedness of individual tree crowns and multi-layered, more densely-packed canopies and hence a more complex 3D-structure. We have tried to illustrate this definition in a new figure 1 of the manuscript. Here, we make use of a widely used mathematical concept to quantify structural complexity, namely the fractal dimension (see response to the reviewer’s next point and e.g. Ehbrecht et al. 2017, Mandelbrot 1975, McGarigal & Marks 1995, Seidel 2018).

- In order to keep the introductory paragraphs more compelling, we avoided to go deeper into further explanations. Instead, we have added a further section to the *Supplementary Information*, introducing and discussing how the TLS measurements were used to derive the components of forest structural complexity.

On the other hand, the description of the SSCI in the Methods is very technical, and it's hard to imagine what the fractal dimension means and how it is related to more traditional measures such as the (horizontal and vertical) variability of functional traits, or variability of height or gap distribution.

- The fractal dimension is a mathematical description of shape complexity and was first introduced by the mathematician Benoit Mandelbrot (see Mandelbrot 1982, *The fractal geometry of nature*). In Mandelbrot (1975) *Stochastic models for the Earth's relief, the shape and the fractal dimension of the coastlines*, a nice example of what the fractal dimension quantifies is introduced, by applying the concept to island shapes. The application in ecology is not new. E.g. Mc Garigal and Marks (1995) used it to quantify landscape structures, Seidel (2018) used it to quantify the structural complexity of individual trees. The SSCI metric is introduced and explained in detail in Ehbrecht et al. (2017). For further clarification, we kindly refer to our response to the reviewer's questions on how SSCI is affected by occlusion. We try to provide a detailed explanation of how the index works there.

Also, if water availability is the major driver of structural complexity, then it would be crucial to include soil variables in the analyses. There are some global soil data sets, like SoilGrids, that could be used for that

- In our revised version of the paper, we now include soil variables in the analysis (see Line 128 – 130, Line 145 – 153, Line 510 – 518).

Second, I think it is problematic to extrapolate the findings to a global map. The authors provide leave-one-out-cross-validation. However, this kind of "validation" is only as good as the sampling used. If there are no plots for example in the Amazon or in central Europe, then there is no way to judge the uncertainties or quality of the extrapolation to those areas.

- As the reviewer has pointed out, a leave-one-out-cross-validation is only as good as the sample, which is why we weren't too confident in just a simple LOOCV. Hence, we have additionally excluded entire biomes from the analysis and thereby up to 30 % of data points, with no substantial reduction in the explanatory power of our model (see Table 2 in manuscript). As mentioned in our cover letter, we have now included a further tropical forest site from Fiji Islands in Oceania, which, surprisingly, did not affect the model outcomes. The average difference in SSCI (after adding a further site) was 0.00123, which further underlines the robustness of our findings.
- Our model predicts a mean SSCI of 5.6 for Central European Mixed Forests and a mean SSCI of 6.57 for Western European Broadleaf Forests. These predictions are well in line observations from unmanaged European beech forests in Central Europe (boxplot on the right). However, we did not include this data in our global study, because the unmanaged plots were managed until 50-60 years ago. We preferred to be more conservative in our selection of study sites, even though Sabatini et al. (2018) classified these sites as old-growth, primary forest.

Figure 2. SSCI of differently managed European beech forests in Hainich-Dün, Germany (Ehbrecht, 2018). Even-aged stands represent different developmental stages in shelterwood systems (thicket, pole woods, immature and mature timber stands). Uneven-aged stands represent stands managed under single tree selection systems with higher tree size heterogeneity. Unmanaged stands represent formerly managed stands where management was ceased about 50-60 years ago. These sites are classified as old-growth forest by Sabatini et al. (2018) and are part of the UNESCO world heritage sites *Ancient and Primeval Beech Forests of the Carpathians and Other Regions of Europe*.

The forest structure is quite different between Borneo and the Amazon, with huge variability within the regions. For example, I would expect a higher structural complexity in the eastern Amazon than in the west. Also, the tallest trees in the Amazon are found in the north-east (>70 m), which should indicate high structural complexity with multi-layered canopies and emergent trees 60-90 m tall, see: Gorgens, E., Nunes, M. H., Jackson, T., Coomes, D., Keller, M., Reis, C. R., ... Ometto, J. P. (2020). Resource availability and availability and disturbance shape maximum tree height across the Amazon University of Helsinki. BioRxiv. <https://doi.org/10.1101/2020.05.15.097683>

- We fully agree that forests of the Amazon and Borneo, or the Congo Basin, are different ecosystems with different structural characteristics. However, differences in specific structural characteristics, such as height of the tallest trees, do not necessarily imply that they are significantly different in terms of their complexity.
- Still, as you have pointed out, emerging canopy trees increase the number of canopy layers, which may increase structural complexity. Our sites in Danum Valley and Maliau Basin in Malaysia (see Shenkin et al. 2019), which are characterized by emerging canopy trees of up to 100 m, also have the highest SSCI values we have measured (among our Fiji- and Valdivian Rainforest site in Chile). The number of canopy layers is accounted for in our structural complexity index by the ENL-component (see our detailed comments on the reviewer's questions related to how SSCI works).
- On the contrary, our Douglas Fir sites in Oregon (USA) are also characterized by tall trees of up to 80m, but their structural complexity is lower. This is probably because conifer trees are characterized by a lower crown plasticity and may not use canopy space as efficiently as broadleaf trees (Jucker et al. 2015). A lower structural complexity of conifer forests compared to broadleaved forests is in line with the findings of Juchheim et al. (2020).
- The huge variability within the regions is reflected in the large variance shown in the violinplots for Indomalayan and Neotropical forests in Figure 3 in the manuscript.

Gorgens et al. 2020 found that wind speed and soil clay content play important roles besides precipitation and radiation. I would assume that wind and soil (eg the combination of shallow soils and strong winds) are also important drivers of forest structure and structural complexity, additionally to the factors the authors considered.

- Thanks for pointing us to the paper by Gorgens et al., of which we were not aware of. As mentioned earlier, we have now included soil data in our analysis. We found significant positive correlations between clay content and SSCI, and negative correlations between sand content and SSCI (see figure 3 in this document). As the reviewer has pointed out in a previous comment, it is crucial to include soil variables in the analysis, if water availability is a major driver of structural complexity. Hence, we decided to use field capacity (water holding capacity) in our analysis instead of soil texture variables. Also, the amount of organic matter and clay minerals play a crucial role for nutrient availability. To account for that, we used the cation exchange capacity and nitrogen content during re-analysis of the dataset.

Figure 3. Relationships between SSCI and soil clay (%), sand (%) and silt (%) (0 - 100 cm soil depth, soilgrids.org, Hengl et al. 2015)

- We have not thought about effects of wind on structural complexity yet. Using the data provided by the Global Wind Atlas, we did not find effects of mean wind speed or mean power density. Probably, we rather need data on the frequency and intensity of storm events to better address this issue. Still, our sampling design isn't really appropriate to test whether the combination of shallow soils and wind affects SSCI, since we did not really cover a soil depth gradient.

Figure 4. No relationships between mean wind speed (m s^{-1}) and mean power density (W m^{-2}). Wind data was derived from globalwindatlas.info (“Global Wind Atlas 3.0, a free, web-based application developed, owned and operated by the Technical University of Denmark (DTU). The Global Wind Atlas 3.0 is released in partnership with the World Bank Group, utilizing data provided by Vortex, using funding provided by the Energy Sector Management Assistance Program (ESMAP). For additional information: <https://globalwindatlas.info>”)

The relationship between structural complexity and height is reported to be hump-shaped (Figure SI2), but this relationship seems to be strongly driven by the Rockefeller forest, whereas tropical forest canopies with tree heights over 60 m seem to be lacking.

- In fact, our tropical site in Danum Valley in Malaysia is characterized by the tallest tropical trees discovered so far (see Shenkin et al. 2019). Our plots in Maliau Basin in Malaysia also include trees with heights above 80 m. However, such tall, emergent canopy trees are not found on each single plot. Since the data points shown in our figures represent site means (hence the standard error bars), the reported canopy heights are somewhat lower.

The Rockefeller forest is a coastal redwood forest with some of the tallest trees on Earth, which is not necessarily the most representative site for temperate conifer forests (for example assuming that those findings would be transferable to temperate conifer forests in Europe or Russia).

- We agree that Rockefeller forest is not representative of a temperate conifer forests in Central Europe. Comparing SSCI values of temperate conifer forests in our global study to mature Norway spruce and Scots pine stands in Central Europe shows that the latter are characterized by slightly lower levels of structural complexity (figure 5). The somewhat lower SSCI of mature Central European Norway spruce forests can be attributed to the type of management and the low tree size heterogeneity in managed forests compared to old-growth, primary forests. Unfortunately, old-growth, primary conifer forests are almost impossible to find in Central Europe.
- It might be surprising that Coastal Redwood forests with their immense size are characterized by a relatively low structural complexity. Coastal Redwood forests are not necessarily characterized by a high tree size diversity, a greater connectedness of individual tree crowns and a high degree of biomass or foliage distribution in three-dimensional space, which is in line with our and Gough et al.’s (2020) definition of forest structural complexity. On the contrary, old-growth Douglas Fir forest in Oregon (US) are significantly more complex than Coastal Redwoods or managed Central European conifer forests, as the above mentioned criteria of structurally complex forests (according to our definition) pertain.

Figure 5. SSCI of old-growth, primary Coastal redwood and Douglas fir forests in North America in comparison to managed Norway spruce and Scots pine stands in Central Europe (data of Norway spruce and Scots pine stands is published in Ehbrecht et al. 2017). Different letters indicate significant differences in mean SSCI (ANOVA, $p < 0.05$).

Therefore, I think an extrapolation to a global map without taking into account additional potential explanatory variables and without having any samples in the Amazon, Asia, Russia or Australia does not seem reasonable. The discussion is well written and addresses some of the issues. I think the study would still be highly relevant and more plausible, if the findings are reported just for the regions that have been sampled.

- Not having a geographically more representative sampling is definitely a limitation of our study. Due to logistic constraints we were not able to cover further sites across the world.
- Still, the robustness of our findings is underlined by the following facts:
 - Removing up to 30% of study sites from the model did not substantially reduce its explanatory power.
 - Adding a further site from a further realm (Oceania) did not change the model outcomes.
 - Soil variables did not improve the explanatory power of our model.
- From a modelling point of view, we argue that having further sites in (1) regions with higher or lower annual precipitation and precipitation seasonality, and (2) sampling forests on different soil types along the climatic range covered would be a more appropriate way to capture more of the global variability of forest structural complexity than having a broader biogeographic coverage. Still, despite the low sample size, we are able to explain almost 90 % of variance in SSCI.
- We decided to keep the global map in the revised version of the manuscript. To take the reviewer's concerns into account, we have added a further map to Figure 4 that shows the uncertainties in model predictions (95% confidence interval) and outlines regions for which we cannot reliably quantify the uncertainty.

- We disagree that reporting the findings only for the regions that have been sampled is plausible. As the reviewer mentioned earlier, the variability within certain regions like the Amazon or Borneo is huge. This large variability is, for example, reflected in the large variance shown in the violin plots for Neotropical and Indomalayan forests (Figure 3 in the manuscript). Hence, even by having a site in the Amazon (or Russia, Asia or Australia), we would not be able to capture that variability. Capturing the variability within a region (compared to an ‘across biomes’-approach) would require a specific sampling design for each specific region that takes different soil conditions and forest ecoregions (see Olson et al. 2001) within each geographic region into account.

Figure 6. 95 % confidence interval of $SSCI_{pot}$ model predictions for current climates. Regions outside the climatic range studied and regions with different soil conditions than our study sites are marked in light blue and yellow, respectively, because we cannot reliably quantify the uncertainty of model predictions for those areas.

Third, how do the authors validate structural complexity and the influence of occlusion using their TLS measurement approach? Wilkes recommend about 20 m distance between measurements to optimize the scan patterns, and Schneider et al. 2019 found that even with a very dense scan pattern of about 10-15 m and 89 scan locations to cover 1 ha of Borneon rainforest, there still remains about 30-90% occlusion in the upper canopy:

Wilkes, P., Lau, A., Disney, M., Calders, K., Burt, A., Gonzalez de Tanago, J., ... Herold, M. (2017). Data acquisition considerations for Terrestrial Laser Scanning of forest plots. *Remote Sensing of Environment*, 196, 140–153. <https://doi.org/10.1016/j.rse.2017.04.030>

Schneider, F. D., Kükenbrink, D., Schaepman, M. E., Schimel, D. S., & Morsdorf, F. (2019). Quantifying 3D structure and occlusion in dense tropical and temperate forests using close-range LiDAR. *Agricultural and Forest Meteorology*, 268(December 2018), 249–257.

<https://doi.org/10.1016/j.agrformet.2019.01.033>

The attached figure shows how occlusion increases with the reduction of scan positions within a 1 ha forest plot in Borneo, with about 40% occlusion when only 6 scan positions are used. With only 5 positions spaced 40 m apart, I wonder how the authors are capable of characterizing tropical forest structural complexity. It seems to me that there might be a systematic under-sampling of the upper canopy, with potentially large parts that remain unexplored/occluded. How does the method perform in tropical forests or in a Redwood forest, where the sampling of the upper canopy is very

challenging? Do the authors have a way to quantify/report occlusion or a way to validate the structural complexity? Also, it seems that the calculation of the SSCI is even based on single scans, which is very problematic in tropical rainforests. Usually, only by adding many different observation positions together, one can get a clearer picture of the overall structure of the forest. A single scan has very limited reach in dense tropical forests (and probably even less with a phase shift vs. a time of flight scanner), and would mostly tell something about the structure of the understory and midstory.

- In order to validate the SSCI, we correlated index values with other metrics of forest structural complexity. Kindly see figure 1 in this document in our response to comments by reviewer 1, where we show how SSCI correlates with the Structural Complexity Index by Zenner & Hibbs (2000), the commonly used Gini-Coefficient of tree diameters (see Lexerød & Eid 2006) and the tree size differentiation index (see Pretzsch 2009). This validation is based on temperate forest plots in Central Europe.
- We do not have a similarly large dataset available for just tropical forests, but for 26 plots along a tropical land-use gradient (oil palm plantations, rubber plantations, jungle rubber and logged tropical rainforest, for details on study sites see Drescher et al. 2016). Unfortunately, we do not have the raw data at hand to compute the above-mentioned tree attribute-based measures of forest structural complexity for comparison with SSCI. Instead, figure 7 shows the relationship between tree size variability (as measured by the coefficient of variation of tree diameters) and SSCI along a land-use intensity gradient.

Figure 7. Relationship between SSCI and the coefficient of variation of tree diameters along a land use-intensity gradient in Sumatra, Indonesia (based on data published in Zemp et al. 2019).

- The development of the SSCI was initially inspired by the idea to develop a single-scan based on index of forest structural complexity that overcomes the issues of occlusion. SSCI is based on two components. One is the fractal dimension of cross-sectional polygons, which acts as a scale-independent, mathematical quantification of complexity. The other one is the *Effective number of layers* (ENL), which is a measure of vertical stratification or layering, and is used to scale the fractal dimension. ENL is basically based on the concept of *foliage height diversity*, which dates back to the seminal work of MacArthur and MacArthur (1961), and was introduced in Ehbrecht et al. (2016).
- To test effects of occlusion on ENL, we compared ENL values between point clouds based on single scans and multiple scans with occlusion of less than 1% (based on a voxel size of 20cm side length. Ray tracing was used to determine the share of occluded voxels, around 40 scans

on a 40 x 40 m plot). We found that ENL can be quantified based on single scans with an RMSE of 9.82% (see figure 8). The slope of the regression line, however, was not significantly different from 1 (i.e. no significant deviation from the 1:1 line).

Figure 8. Comparison of ENL derived from point clouds based on single-scan and multiple scan approach. Comparison is based on 25 forest plots of 40 x 40 m in size. Each plot was scanned with at least 40 scans to minimize the share of occluded voxel to around 1%. Forest plots were selected to cover a gradient of forest structural complexity, ranging from single-layered even-aged conifer stands to structurally complex old-growth European beech stands. Source: Ehbrecht et al. (2016).

- The fractal dimension quantifies the shape complexity of cross-sectional polygons, which are based on the hemispherical view of the scanner. A higher forest structural complexity results in a more complex shape of the cross-sectional polygons (see figure 9 and Ehbrecht et al. 2017, Ehbrecht 2018). Even though large parts of the upper canopy remain occluded, single laser beams still traverse through the 3D space through tiny gaps and detect objects in upper canopy layers, which creates spikes in the cross-sectional polygons and increases fractal dimension values. So, the more heterogeneous the foliage or biomass distribution in three-dimensional space, the more irregular and complex the shape of the polygon, the higher the fractal dimension values and thus structural complexity. However, as mentioned earlier, the fractal dimension itself is scale-independent. I.e. that a dense thicket may be characterized by similar fractal dimension values as a multi-layered, tall tropical forest. This is where ENL comes into play to scale the fractal dimension values of the cross-sectional polygons, by using the natural logarithm of ENL as an exponent.
- With regard to the paper by LaRue et al. (2019) the reviewer has mentioned earlier, the fractal dimension of cross-sectional polygons quantifies the *internal heterogeneity* that LaRue et al. describe.

Figure 9. Graphical visualization of three-dimensional point clouds, cross-sectional polygons and vertical profiles of a European beech forests with low (A, SSCI = 4.3) and high forest structural complexity (B, SSCI = 7.1). Modified after Ehbrecht et al. (2017). Images of three-dimensional point clouds are based on a dataset published in Juchheim et al. (2017).

- The findings of this study and also from other studies (e.g. Zemp et al. 2019), suggest that SSCI quantifies the structural complexity of tropical forests appropriately. First, the SSCI values we measured in tropical forests, such as Danum Valley and Maliau Basin in Malaysia, as well as in tropical forests in Fiji, were in fact the highest values we have ever measured (except for a temperate rainforest in Chile). Second, SSCI has been successfully used to distinguish the structures of differently managed forests compared to old-growth forests including logged and primary tropical forest (see figure 10 as well as Ehbrecht et al. 2017, Stiers et al. 2018, Juchheim et al. 2020)

Figure 10. SSCI of differently managed temperate broadleaf, temperate conifer and tropical moist broadleaf forests in comparison to primary forests. Even-aged stands represent shelterwood systems (for European beech stands in Europe) or plantations (for Douglas fir stands in the US and Scots pine stands in Europe). Uneven-aged E. beech stands represent stands managed under single tree selection (Europe). Unmanaged E. beech stands were formerly managed until 50-60 years ago. Data of logged *Dipterocarpaceae* forests was collected in Jambi Province, Sumatra, Indonesia (see Zemp et al. 2019). Data of primary European beech (Ukraine), Douglas fir (Oregon, US) and *Dipterocarpaceae* (Malaysia) forests was collected within the frame of this study.

- Ultimately, this is probably not the kind of validation the reviewer requests for tropical forest. Still, we hope that our explanations on how the index works and is constructed, how it correlates with other measures of structural complexity, as well as its capability to differentiate differently managed forests from primary forests, increases the reviewer’s confidence that SSCI quantifies what it is supposed to quantify. It has been used in numerous studies so far (see methods section).

In conclusion, I would recommend to focus on the conclusions that can be directly drawn from the sites measured, for example moving supplementary figure 1 to the main text to show the distribution of the plots and representativeness of climate regions, and move the global maps to the supplementary as “experimental results”, unless a validation can be provided for the regions that were not sampled. I would also expect some discussion of the measurement approach, including potential limitations, and possible demonstration of the applicability of the approach in tropical rainforests.

- To further discuss our measurement approach, we have now included parts of our explanations on how SSCI works and has been validated in the SI. We found it more compelling to move methodical discussions to the SI.
- As mentioned in our response to a previous comment, we decided to keep the global maps and now try to acknowledge their limitations in more detail (Line 327 – 338, 372 - 409). In this context, we have added a map of uncertainty of model predictions to figure 4 in the manuscript. In this map, we also highlight areas for which we cannot reliably quantify the uncertainty of model predictions (Line 234 – 237).
- We hope our explanations on why we think our findings are robust and allow for an extrapolation to a global map increased the reviewer’s confidence in our upscaling approach.

Two detailed comments:

L255-258: But see Harrison et al. 2020 for a recent reference:

Harrison, S., Spasojevic, M. J., & Li, D. (2020). Climate and plant community diversity in space and time. Proceedings of the National Academy of Sciences of the United States of America.

<https://doi.org/10.1073/pnas.1921724117>

- Thanks for the link to the paper, of which we weren't aware. We have taken it into account in the revised version.

L266: But there might also be complex feedbacks, for example frequent natural fire disturbance can lead to increased local forest structure variability that make forests more resilient against tree mortality by fires, see:

Koontz, M. J., North, M. P., Werner, C. M., Fick, S. E., & Latimer, A. M. (2020). Local forest structure variability increases resilience to wildfire in dry western U.S. coniferous forests. Ecology Letters, ele.13447. <https://doi.org/10.1111/ele.13447>

- We agree that there also might be complex feedbacks. However, please note that site-scale variability of forest structural complexity is not within the focus of this study.

Reviewer #3

We thank reviewer 3 for the inspiring suggestions to improve the manuscript. The most important changes to the manuscript in relation to the comments by reviewer 3 are:

- Introducing the biotic and abiotic determinants in more detail (see new Figure 1)
- Including and discussing further literature in relation to other data sources (GEDI-LiDAR data) and species-specific responses to climate change.
- Inclusion of soil variables in a comprehensive re-analysis of the dataset.

The goal of the paper is

To understand the future impacts of climate change on the functioning of forest ecosystems and biodiversity, we urgently require knowledge about forest structural complexity in the context of climate change scenarios.

The paper in question contributes to a better understanding of the global patterns and climatic drivers of forest structural complexity in various biomes. Local patterns of forest structures are determined from extensively conducted terrestrial LiDAR measurements, which are transferred to complex global patterns and forecasts that are made about their future reactions and changes in the face of climate change.

The investigation of the relationship between water availability and the diversity of functional characteristics with a particular focus on the structural, morphological and physiological characteristics is very well structured, has been very well statistically analysed and is written well. The paper in question therefore provides a valuable and important methodological contribution to investigate the complexity of biodiversity and climate change.

The paper should be accepted, if the following critical factors are discussed in more detail.

The paper uses globally modelled climate data from 1971-2000 from the WorldClim2-Database to test the relationships between climate variables and forest structural complexity, canopy height, area and the openness of the canopy. In principle I am not sure whether the interlinked scales – globally modelled 1km raster data on the climate and terrestrial LiDAR-data - TLS (ca. 1m) can be placed in relation to one another.

- The WorldClim2-Database offers spatially explicit climate data with the spatially highest resolution currently available and is hence the best available dataset to get estimates of climate conditions for our study sites. If we understand reviewer correctly, the assumption is whether climate data that had been measured directly at each site had been a better choice to relate climate and forest structural complexity? Such data was only available for some of our sites. In order to be consistent with the data sources, we found the WorldClim2-Database to be the best choice to have estimates of annual precipitation and precipitation seasonality for each site.
- Furthermore, we think the interlinked scales can be placed in relation to one another, because the plots per site were approximately distributed across 1 km². To get a robust estimate of site-scale structural complexity, we averaged the ~15 plots per site to a mean SSCI and provided the standard error. As such, we do not compare single terrestrial laser scans with a spatial resolution of ca. 1m with modelled climate data of 1 km resolution.

A combination of TLS with air- and spaceborne LiDAR (GEDI-3D LiDAR) would have given the study added value (even if only conducted on specific test sites) and would have enabled important statements regarding scale dependency and comparability.

- We agree that integrating also air- or spaceborne LiDAR would have given the study added value. Unfortunately, coordinating airborne LiDAR missions for each of the 20 sites was beyond our possibilities and funding. Spaceborne LiDAR is now available, but is subject to a much coarser resolution than TLS. We question whether the resolution of GEDI-3D spaceborne LiDAR is sufficiently high to quantify forest structural complexity at the plot level. However, using TLS for ground-truthing of spaceborne LiDAR is definitely an important topic in the next years.
- To address these concerns, we now specifically address the GEDI-mission in our discussion (Line 349 – 355). Despite current issues in the comparison of TLS and spaceborne-data due to different resolutions, which will certainly be solved, it may enable to map the actual structural complexity of the world's forests. Relating the actual to the potential structural complexity would help to better interpret the current state of forests worldwide and to improve the identification of intact of forest landscapes of high-conservation-value.

Throughout the entire paper, reference is primarily given to the structural complexity of trees in the context of various climate variables (in particular water availability). Please discuss other important factors like for example the argument that reactions to climate change are species-specific (Walther, G.; Post, E.; Convey, P.; Menzel, A.; Parmesan, C.; Beebee, T.J.C.; Fromentin, J.; I, O.H.; Bairlein, F. Ecological responses to recent climate change. *Nature* 2002, 416, 389–395.)

- In order to avoid misunderstandings: This study focuses on forest structural complexity at the stand level and not at the level of individual trees.
- Still, the reviewer addresses an interesting point we have not thought about yet. In each biome, different tree species may respond differently to climate change. Thus, climate change induced changes in forest structural complexity may differ between forests of different species compositions and diversity within the same biome. We have now tried to address this topic in the discussion and take the proposed reference into account (Line 387 – 392).

Geodiversity considerably shapes biodiversity. In this context, it is the climate and the water regimes

that are the strongest drivers, as you confirmed in your paper. However, the paper provides very little reference to other important abiotic factors of geodiversity such as soil moisture, geomorphology or as mentioned above species dependency.

- We fully agree that a multitude of factors shape forest structural complexity. To better address the reviewers concerns, we have revised figure 1 in the manuscript, revised part of the introduction (Line 93 – 98) and discuss the determinants of forest structural complexity in more detail from Line 289 – 315.
- As requested also by the other reviewers as well, we have now included soil data in our analysis (see Line 128 – 130, Line 145 – 153, Line 510 – 518). We found significant effects of water holding capacity (field capacity) and soil nitrogen on SSCI. Still, these did not improve our model based on annual precipitation and precipitation seasonality.
- Moreover, we have revised the introduction, including figure 1, and introduce the manifold determinants of forest structural complexity in more detail and try to outline more why climate may shape forest structural complexity by controlling its direct determinants.

In the paper at hand it is emphasized how the climate-structure-relationship is driven by climatic effects (water availability) on the functional characteristics of plants such as leaf area density, biomass and canopy height. However, structural vegetation characteristics/ growth characteristics are controlled by factors such as light availability and intensity and photosynthesis. These factors can easily be ascertained from TLS. Please explain why you decided not to include these geo-factors in your analysis.

- In our analysis, we used solar radiation during the growing season (derived from the WorldClim2-Database) as measure of light availability. However, it did not predict forest structural complexity as good as water availability related variables.
- With regard to derive measures of light availability from TLS: Deriving canopy openness has actually shown to be a good proxy for light availability beneath the canopy as it is strongly correlated with the diversity of ground vegetation (Dormann et al. 2020). However, using it as a measure of light availability in our analysis did not work from a statistical point of view, because measurements of forest structural complexity and light availability are not statistically independent, since they are both based on the same 3D point cloud.
- To discuss the mechanisms underlying the correlations between water availability and structural complexity, we have added a further paragraph to the discussion. In that paragraph we now discuss the role of light in more detail.

Detailed comments:

Line 44-46: “Our analyses reveal distinct latitudinal patterns of forest structure and show that hotspots of high structural complexity coincide with hotspots of biodiversity”

Comment: How are the hotspots of biodiversity determined?

- We determined the hotspots based on the dataset provided by Kier et al. (2005). Kier et al. compiled estimates of species richness of vascular plants for each ecoregion as classified by Olson et al. (2001). Since we used the same classification, we compared our model predictions for SSCI with the compilation provided by Kier et al. (2005).
- We now mention how we determined these relationships in the methods section.

Line 52

“yet unclear impacts..”

Comment: Please specify which impacts are meant here.

- We aimed to articulate that the effects of climate changed-induced changes of forest structure and functioning have, so far, unclear impacts on biodiversity and ecosystem functions, i.e. the consequences for biodiversity and functions remain unclear.

Line 56-58

„...individual forest structure attributes, including leaf area, biomass and canopy height..”

Comment: You refer to studies that connect climate dependency and global patterns of individual forest structure characteristics like for example leaf area density, biomass and canopy height. However, you fail to mention the importance of reactions to climate change depending on the tree species (see above comment). Please include this element of species dependency in your paper.

- Okay. We have tried to follow the reviewer’s suggestion and discuss this aspect in the paragraph on the limitations of our climate change induced changes in $SSCI_{pot}$ (Line 397 – 399, 404 – 409).

Line 77

Comment: Please refer to the importance of new LiDAR satellite technologies i.e. GEDI-3D LiDAR in relation to your hypothesis and include this in your paper.

- We have tried to follow the reviewer’s suggestion. However, we thought that this point may fit better in the discussion and now discuss the importance GEDI-3D LiDAR data in relation to our global map (Line 362 – 368).

From Line 84

“...sound understanding of the biotic and abiotic drivers ...”

Comment: Here you address the significance of biotic and abiotic drivers in the context of the structural diversity of forests, but then you only mention biotic drivers. Please be more explicit about the abiotic drivers (e.g. geodiversity, except for the climate, which is already included) and cite the respective literature.

- We have revised the introduction and now introduce the determinants of forest structural complexity in more detail. In this context, we have also revised figure, which now includes a scheme of how abiotic and biotic factors drive forest structural complexity (Line 103 – 106).
- We have tried to be more explicit about the abiotic drivers by discussing their relevance in the discussion (Line 262 – 289).

From Line 230

„...standardized global field campaign

Comment: Please be more explicit about the concept of the „standardized global field campaign“.

- We have tried to make it more intuitively understandable and now just speak of “global field campaign”. “Standardized” referred to the fact that we used the same method and sampling design at each site.

From Line 298

„However, actual structural complexity is subject to the temporal and spatial dynamics of changes in species composition and disturbance regimes,....”

Comment: The authors name the greatest deficits of the paper here. Here it should be explicitly mentioned which approaches are currently available to monitor or model the temporal and spatial dynamics of forest ecosystems (e.g. linking LiDAR measurements and forest growth models).

Minimising the discussion here to soil texture or nutrient availability is not expedient, as both of these factors are governed by smaller geodynamics.

- We now explicitly mention the importance of GEDI 3D- LiDAR for monitoring and mapping the spatio-temporal dynamics of forest ecosystems in the discussion. Linking LiDAR measurements with forest growth models is certainly an important point. However, we found it more appropriate to focus the discussion more on how our global maps of the ‘potential’ structural complexity may serve as benchmark for mapping the ‘actual’ structural

complexity. Relating the actual structural complexity to the potential structural complexity could indicate the state of intactness of forest landscapes (Line 362 – 368).

From Line 316-336

Comment: The authors discuss the prediction accuracy of the potential structural complexity regarding the climate models used. These aspects of the discussion are very good but they also point out the shortcomings of the study, namely that the local complexity of forest ecosystems was determined, but that complexity is not a spatio-temporal reflection of disturbance regimes. Furthermore, the comparability of the scales must be questioned (climate variables, variables of complexity from the TLS). The value of a future SSCI and its relative changes can thus not really be placed in the context of disturbance regimes / climate change.

- We are not exactly sure what is meant with “...,but that complexity is **not** a spatio-temporal reflection of disturbance regimes”. We determined the local complexity of old-growth, primary forests, i.e. late-successional forests with no signs of recent (larger-scale) disturbances, except for small-scale disturbance like single tree-falls gaps.
- Future SSCI is not meant to reflect actual changes in structural complexity by 2070, but rather the changes in the potential for certain levels of structural complexity. We now try to highlight that further in the discussion of future SSCI and also in the figure caption (Line 238 – 243, 372 – 388).

Line 450:

Trabucco & Zomer (2018) – Literature reference is missing.

- We have added the missing literature reference.

Methods

The methods section was written exceptionally well and in detail. If the paper needs to be shortened in any way then it would make sense to move some of the methods section to the appendices.

- Thanks. Since there were no requests to shorten the paper, we left the methods section as it is.

References

Dormann, C. F., Bagnara, M., Boch, S., Hinderling, J., Janeiro-Otero, A., Schäfer, D., ... & Hartig, F. (2020). Plant species richness increases with light availability, but not variability, in temperate forests understorey. *BMC ecology*, 20(1), 1-9.

Drescher, J., Rembold, K., Allen, K., Beckschäfer, P., Buchori, D., Clough, Y., ... & Irawan, B. (2016). Ecological and socio-economic functions across tropical land use systems after rainforest conversion. *Philosophical Transactions of the Royal Society B: Biological Sciences*, 371(1694), 20150275.

Ehbrecht, M. A. (2018). Quantifying three-dimensional stand structure and its relationship with forest management and microclimate in temperate forest ecosystems. *Dissertation, University of Göttingen* <http://hdl.handle.net/11858/00-1735-0000-002E-E341-A>

Ehbrecht, M., Schall, P., Juchheim, J., Ammer, C., & Seidel, D. (2016). Effective number of layers: a new measure for quantifying three-dimensional stand structure based on sampling with terrestrial LiDAR. *Forest Ecology and Management*, 380, 212-223.

- Ehbrecht, M., Schall, P., Ammer, C., & Seidel, D. (2017). Quantifying stand structural complexity and its relationship with forest management, tree species diversity and microclimate. *Agricultural and Forest Meteorology*, 242, 1-9
- Gough, C. M., Atkins, J. W., Fahey, R. T., Hardiman, B. S., & LaRue, E. A. (2020). Community and structural constraints on the complexity of eastern North American forests. *Global Ecology and Biogeography*.
- Juchheim, J., Ehbrecht, M., Schall, P., Ammer, C., & Seidel, D. (2020). Effect of tree species mixing on stand structural complexity. *Forestry: An International Journal of Forest Research*, 93(1), 75-83
- Jucker, T., Bouriaud, O., & Coomes, D. A. (2015). Crown plasticity enables trees to optimize canopy packing in mixed-species forests. *Functional Ecology*, 29(8), 1078-1086.
- Lexerød, N. L., & Eid, T. (2006). An evaluation of different diameter diversity indices based on criteria related to forest management planning. *Forest Ecology and Management*, 222(1-3), 17-28.
- MacArthur, R. H., & MacArthur, J. W. (1961). On bird species diversity. *Ecology*, 42(3), 594-598.
- McGarigal, K., & Marks, B. J. (1995). FRAGSTATS: spatial analysis program for quantifying landscape structure. *USDA Forest Service General Technical Report PNW-GTR-351*
- Mandelbrot, B. B. (1975b). Stochastic models for the Earth's relief, the shape and the fractal dimension of the coastlines, and the number-area rule for islands. *Proceedings of the National Academy of Sciences*, 72(10), 3825-3828.
- Mandelbrot, B. B. (1982). The Fractal Geometry of. *Nature*, 394-397.
- Seidel, D. (2018). A holistic approach to determine tree structural complexity based on laser scanning data and fractal analysis. *Ecology and evolution*, 8(1), 128-134
- Olson, D. M., Dinerstein, E., Wikramanayake, E. D., Burgess, N. D., Powell, G. V., Underwood, E. C., ... & Loucks, C. J. (2001). Terrestrial Ecoregions of the World: A New Map of Life on Earth A new global map of terrestrial ecoregions provides an innovative tool for conserving biodiversity. *BioScience*, 51(11), 933-938.
- Pretzsch, H. (2009). Forest dynamics, growth, and yield. In *Forest dynamics, growth and yield* (pp. 1-39). Springer, Berlin, Heidelberg.
- Kier, G., Mutke, J., Dinerstein, E., Ricketts, T. H., Küper, W., Kreft, H., & Barthlott, W. (2005). Global patterns of plant diversity and floristic knowledge. *Journal of Biogeography*, 32(7), 1107-1116
- Sabatini, F. M., Burrascano, S., Keeton, W. S., Levers, C., Lindner, M., Pötzschner, F., ... & Debaive, N. (2018). Where are Europe's last primary forests?. *Diversity and Distributions*, 24(10), 1426-1439.
- Schall, P., Schulze, E. D., Fischer, M., Ayasse, M., & Ammer, C. (2018). Relations between forest management, stand structure and productivity across different types of Central European forests. *Basic and Applied Ecology*, 32, 39-52
- Seidel, D. (2018). A holistic approach to determine tree structural complexity based on laser scanning data and fractal analysis. *Ecology and evolution*, 8(1), 128-134
- Shenkin, A., Chandler, C. J., Boyd, D. S., Jackson, T., Disney, M., Majalap, N., ... & Wilkes, P. (2019). The world's tallest tropical tree in three dimensions. *Frontiers in Forests and Global Change*, 2, 32.

Stiers, M., Willim, K., Seidel, D., Ehbrecht, M., Kabal, M., Ammer, C., & Annighöfer, P. (2018). A quantitative comparison of the structural complexity of managed, lately unmanaged and primary European beech (*Fagus sylvatica* L.) forests. *Forest Ecology and Management*, 430, 357-365.

Wieder, W. R., Boehnert, J., Bonan, G. B. & Langseth, M. RegridDED Harmonized World Soil Database v1.2. *ORNL DAAC* (2014) doi:<https://doi.org/10.3334/ORNLDAAAC/1247>

Zemp, D. C., Ehbrecht, M., Seidel, D., Ammer, C., Craven, D., Erkelenz, J., ... & Kreft, H. (2019). Mixed-species tree plantings enhance structural complexity in oil palm plantations. *Agriculture, Ecosystems & Environment*, 283, 106564.

Zenner, E. K., & Hibbs, D. E. (2000). A new method for modeling the heterogeneity of forest structure. *Forest ecology and management*, 129(1-3), 75-87

Reviewer Comments, second round -

Reviewer #2 (Remarks to the Author):

First of all, the authors have to be lauded for the tremendous work they put into the revisions and the very detailed, thorough and helpful responses to the reviews. The manuscript was much improved! The additional map of model confidence and marked areas out of the predictable range is critical, and helps a lot to put the structural complexity map in context. The additional explanations and discussions of the index and the maps are very helpful, important and improved the manuscript overall.

The additional explanations about the structural complexity index have been very helpful too, and I see the value of this approach. I agree that the authors put a huge effort in measuring field plots globally and that adding additional sites at this point is not possible. Various teams are working on forest TLS around the globe, and efforts are under way to combine efforts in a central TLS database, which would help to improve this situation for future studies. I would therefore encourage the authors to consider publishing the TLS datasets (point clouds).

A few more considerations came to my mind regarding the index and the interpretation of the maps:

Thinking of general size structures of forests (eg Weibull distributions of DBH), then small trees are very abundant and large trees get increasingly rare. The variability of DBHs (without abundance weighting) seems therefore more strongly driven by the variability of small to mid-sized trees that are most abundant. Similarly, I get the impression that the SSCI is more strongly reacting to the complexity of the understory and mid-story, than for example a few rare big trees (Therefore, a relatively good correlation with CV of DBH). Rare big trees seem to have an "unproportionally" (with regard to abundance) large influence on total biomass though, and potentially also "external" complexity such as canopy surface roughness or the canopy volume profile. So to me it seems the index is more strongly related to leaf area than to biomass heterogeneity (Line 69), and to internal complexity as the authors also stated in their response. If those assumptions are correct, I would revise the second paragraph of the introduction to make this clearer.

The new Figure 1 is nice, but needs a few adjustments. Topography and Microclimate are missing here, maybe could be combined with soil. There should be an arrow from Climate to Tree species composition. Natural disturbance could also influence soil (eg land slides, rock falls, river flooding). And I would think that natural and anthropogenic disturbance could alter functional diversity without a change of species composition.

Thank you for the discussion of the SSCI in the context of Europe. I did actually not think of the fact that there are rarely any unmanaged forests in Europe. Therefore, it is difficult to know how the distribution of potential structural complexity across Europe would look like. This reminds me of the study by Thonicke et al. (2020), simulating the potential functional diversity across Europe with an individual-based dynamic global vegetation model. The study is different, in the sense that functional diversity measures based on tree height, SLA, leaf longevity and wood density were used, but the model takes into account competition, certain disturbances including fire, soils, and of course (most importantly in this context) the meteorological drivers and climatic constraints. It is a little bit difficult to see the variation within Europe in the global context, but it seems that there might be some resemblance with the functional divergence map as predicted by the DGVM, which would make sense to be related to structural complexity. Also, I was surprised to see high functional diversity in southern Scandinavia in Thonicke et al, but this is because the climate and environmental conditions there could support mixtures of broadleaf and coniferous species with strongly differing functional traits (and I assume also structural properties). I don't see this in the proposed map, but it would be interesting to hear the authors opinion on the comparison to DGVM modeling results.

Thonicke, K., Billing, M., Bloh, W., Sakschewski, B., Niinemets, Ü., Peñuelas, J., ... Walz, A. (2020). Simulating functional diversity of European natural forests along climatic gradients. *Journal of Biogeography*, 47(5), 1069–1085. <https://doi.org/10.1111/jbi.13809>

A main concern remains with the structural complexity change with climate change. To me it does not seem possible to make such a prediction with the data presented, even if it is a prediction of potential complexity not considering certain natural and anthropogenic disturbances. As was discussed in the reviewer responses, and also added to the discussion of the manuscript, how forest structural complexity will change with climate change depends on the species. Additionally, it depends on the evolutionary history and genetic structure and diversity of the forest communities. Tree species also tend to be plastic, quite adaptable and not shift in range as much as one would think of considering their optimal environmental niche. It seems more likely to see strong shifts in species and functional composition after large-scale disturbances such as insect attacks/forest diebacks, storms or large fires, when forests regrow and assemble in a new way. This is very difficult to predict though, and also depends on the local species and gene pools. Without considering abrupt changes like this, it is still also difficult to predict gradual changes of forest structure with climate change. Species diversity is considered to provide an “insurance” effect, based on the assumption that with a larger species pool it is more likely to have species that can adapt to and be competitive in a new climate or under climate extremes. Therefore, high diversity communities are considered more stable (likely also in structural complexity) and are supposed to adapt better to climate change than low diversity communities. How is this considered in your prediction? In the end, it also really depends on which species might be more vulnerable and die. An experimental study by Bunker et al. (2005) showed that biomass would vary by more than 600% and biological insurance varied by more than 400% among different extinction scenarios, depending on which species die (first). Therefore, I don't think that one can predict the potential change in structural complexity without historical data (or experiments) showing how specific forest communities and species assemblages react to climate change. I don't think one can assume the same response (change of SSCI) for different species assemblages with varying evolutionary history globally to a change in precipitation and precipitation seasonality. Trisos et al. (2020) for example show that certain species might experience unprecedented climatic conditions as early as 2030 and some more until 2050 that are outside their historic climatic niches. How does the prediction of potential SSCI change take into account different species climatic niches, species plasticity and likelihood to migrate or die? You could show predictions of precipitation and/or precipitation seasonality change until 2070, and discuss where we might see positive or negative changes in structural complexity (without estimating the climate effect on SSCI), but the potential “level” for structural complexity to be expected in the future in a region is not just defined by the climate but also by the species pools and evolutionary history. Unless I misunderstand the modeling, I don't think any of this is taken into account when the authors estimated the climate change effect on SSCI.

Bunker, D. E. (2005). Species Loss and Aboveground Carbon Storage in a Tropical Forest. *Science*, 310(5750), 1029–1031. <https://doi.org/10.1126/science.1117682>

Trisos, C. H., Merow, C., & Pigot, A. L. (2020). The projected timing of abrupt ecological disruption from climate change. *Nature*, 580(7804), 496–501. <https://doi.org/10.1038/s41586-020-2189-9>

Reviewer #3 (Remarks to the Author):

I am very satisfied with the revision and incorporation of the comments made.
I recommend accepting the paper in the present form.

Reviewer #2 (Remarks to the Author):

First of all, the authors have to be lauded for the tremendous work they put into the revisions and the very detailed, thorough and helpful responses to the reviews. The manuscript was much improved! The additional map of model confidence and marked areas out of the predictable range is critical, and helps a lot to put the structural complexity map in context. The additional explanations and discussions of the index and the maps are very helpful, important and improved the manuscript overall.

The additional explanations about the structural complexity index have been very helpful too, and I see the value of this approach. I agree that the authors put a huge effort in measuring field plots globally and that adding additional sites at this point is not possible. Various teams are working on forest TLS around the globe, and efforts are under way to combine efforts in a central TLS database, which would help to improve this situation for future studies. I would therefore encourage the authors to consider publishing the TLS datasets (point clouds).

- Thanks a lot for the suggestion. We are very much interested to contribute our TLS point clouds to a central TLS database. We would be grateful for further information on the current progress and options for contribution.
- The algorithm to compute the stand structural complexity index (SSCI) used in this study is now available at <https://github.com/ehbrechtetal/Stand-structural-complexity-index---SSCI>. The link is now also included in the code availability statement in the manuscript. Efforts are under way to make the SSCI-algorithm available as an R-package, so that the wider community can use it more easily for their research.

A few more considerations came to my mind regarding the index and the interpretation of the maps:

Thinking of general size structures of forests (eg Weibull distributions of DBH), then small trees are very abundant and large trees get increasingly rare. The variability of DBHs (without abundance weighting) seems therefore more strongly driven by the variability of small to mid-sized trees that are most abundant. Similarly, I get the impression that the SSCI is more strongly reacting to the complexity of the understory and mid-story, than for example a few rare big trees (Therefore, a relatively good correlation with CV of DBH). Rare big trees seem to have an “unproportionally” (with regard to abundance) large influence on total biomass though, and potentially also “external” complexity such as canopy surface roughness or the canopy volume profile. So to me it seems the index is more strongly related to leaf area than to biomass heterogeneity (Line 69), and to internal complexity as the authors also stated in their response. If those assumptions are correct, I would revise the second paragraph of the introduction to make this clearer.

- We agree that, considering the general size structures of primary forests (e.g. inverse J-shaped diameter distributions), the very high number of very small trees compared to very few or rare big trees have a strong effect on dispersion measures such as the CV of DBH.
- As pointed out in our revision, SSCI strongly correlates not only with a simple CV of DBH, but with other tree attribute-based measures of forest structural complexity as well, such as Zenner & Hibbs' (2000) Structural Complexity Index. The index developed by Zenner & Hibbs is less affected by the abundance of small trees and a few or rare big trees can have a strong impact on index values. In their approach, tree tops are connected to each other using an irregular triangle network approach (e.g. Trilauney triangulation). As a result, big trees increases the surface roughness (i.e. external complexity), which is then expressed in higher index values.
- Indeed, rare big trees have a disproportionate influence on stand biomass (Lutz et al. 2018). However, it is not clear yet to what degree internal decay may reduce their actual importance for

biomass and hence carbon sequestration (Marra et al. 2018). Whether they should have a disproportionate influence on forest structural complexity is an interesting question. The abundance of tall trees is partially reflected in the index component that describes vertical stratification (ENL). An increasing abundance of tall trees would also result in higher ENL values and increase structural complexity.

- We still think that our definition of structural complexity is line with how the index works, because the disproportionate influence of big trees on stand biomass results in a heterogeneous, i.e. uneven distribution of biomass in 3D space.
- Furthermore, foliage is also biomass. Using the term biomass in the definition integrates woody components and foliage, which are not differentiated in the point clouds. Hence, we think that our definition in line 69 is appropriate.

The new Figure 1 is nice, but needs a few adjustments. Topography and Microclimate are missing here, maybe could be combined with soil. There should be an arrow from Climate to Tree species composition. Natural disturbance could also influence soil (eg land slides, rock falls, river flooding). And I would think that natural and anthropogenic disturbance could alter functional diversity without a change of species composition.

- Thanks a lot for this comment. We have taken the reviewer's suggestions into account and revised the figure accordingly. However, we disagree with the reviewer's point on microclimate. Actually, microclimate is rather determined by forest structure (and topography) than being a determinant of structure (see Ehbrecht et al. 2019, Jucker et al. 2018). Topography may control variability in structural complexity on a local scale through differences in soil conditions (e.g. shallow soils on a hill top with rock outcrops, water-logged soils in valleys or depressions) and climate (e.g. higher rainfall on luv-ward side, less rainfall on the lee-ward side of a mountain range). We have now included topography in the figure. However, we think that including microclimate (in addition to further arrows linking disturbances with functional diversity, soil, etc.) rather confuses the actual focus of the figure, since its relationship with structure is indirectly included in the icons for "climate" and "topography and soil".
- The arrow linking climate with species composition was unintentionally not shown. It got lost when converting the figure from PowerPoint to PDF/JPG. We are sorry for this confusion. It is now included in the revised version.

Thank you for the discussion of the SSCI in the context of Europe. I did actually not think of the fact that there are rarely any unmanaged forests in Europe. Therefore, it is difficult to know how the distribution of potential structural complexity across Europe would look like. This reminds me of the study by Thonicke et al. (2020), simulating the potential functional diversity across Europe with an individual-based dynamic global vegetation model. The study is different, in the sense that functional diversity measures based on tree height, SLA, leaf longevity and wood density were used, but the model takes into account competition, certain disturbances including fire, soils, and of course (most importantly in this context) the meteorological drivers and climatic constraints. It is a little bit difficult to see the variation within Europe in the global context, but it seems that there might be some resemblance with the functional divergence map as predicted by the DGVM, which would make sense to be related to structural complexity. Also, I was surprised to see high functional diversity in southern Scandinavia in Thonicke et al, but this is because the climate and environmental conditions there could support mixtures of broadleaf and coniferous species with strongly differing functional traits (and I assume also structural properties). I don't see this in the proposed map, but it would be interesting to hear the authors opinion on the comparison to DGVM modeling results.

Thonicke, K., Billing, M., Bloh, W., Sakschewski, B., Niinemets, Ü., Peñuelas, J., ... Walz, A. (2020). Simulating functional diversity of European natural forests along climatic gradients. *Journal of Biogeography*, 47(5), 1069–1085. <https://doi.org/10.1111/jbi.13809>

- Thanks a lot for the link to the paper by Thonicke et al. (2020), of which we were not aware.
- Links between functional diversity and structural complexity have not been tested so far and this is certainly a very interesting topic that should be addressed in future research.
- Thonicke et al. (2020) have made their maps of functional divergence (FD), functional richness (FR) and functional evenness (FE) available to us so that we could relate these measures to the SSCI. We used a sampling grid of 100 km distance between sample points to sample FD, FR, FE and SSCI_{pot} for each sample point. Please see figure 1 below for correlations between SSCI and functional diversity.
- SSCI increases with increasing functional richness, but saturates at a certain level. This effect is similar to the effect of tree species diversity on SSCI shown in Ehbrecht et al. (2017) or Zemp et al. (2019). Structural complexity increases with increasing tree species diversity due to complementarity in crown architectures. A further increase in tree species diversity can be redundant as additional species do not necessarily increase complementarity in niche/canopy space occupation.

Figure 1. Relationships between measures of functional diversity and potential structural complexity (functional richness (*left*), functional evenness (*center*), functional divergence (*right*)). Data points are based on a sampling grid with a distance of 100 km between sample points ($n = 560$), which was confined to the extents of the maps published in Thonicke et al. (2020). For each sample point, the estimate of potential structural complexity was picked from the global map shown in Figure 4a of the original manuscript, and the values for functional richness, evenness and divergence were derived from the maps by Thonicke et al. (2020). The relationship between functional richness and SSCI was then modelled using a non-linear, asymptotic model (*nls2*-package, $p < 0.001$). The relationship between functional evenness and SSCI was modelled using a linear regression model ($p < 0.001$).

- A decrease in structural complexity with increasing functional evenness might depend on the relationship between functional richness and evenness, as functional evenness may be high in sites with low functional richness (see Figure 2 below).
- Better understanding these relationships requires further investigations and is beyond the scope of this paper. This topic certainly deserves more attention in future research.
- We agree that the variation is not well visible in the figure shown in the manuscript. We would like to make the GEOTiff of our potential structural complexity maps (incl. the uncertainty map) in ~ 1 arcsecond resolution available in conjunction with a publication of the manuscript. That will allow to zoom into Europe (and other parts of the world), which will make it easier to see the

variability within an area of interest. Also, we aim at making the maps available to the wider community for further research. We have uploaded the maps to the data repository hosted by the University of Göttingen and reserved a DOI, which will be published in case of a successful publication of the manuscript (see data availability statement in the manuscript).

Figure 2. Relationship between functional evenness and functional richness in the dataset by Thonicke et al. (2020). The relationship was modelled using a linear model ($\text{lm}(\text{functional evenness} \sim \log(\text{functional richness}))$) and was significant at $p < 0.001$.

A main concern remains with the structural complexity change with climate change. To me it does not seem possible to make such a prediction with the data presented, even if it is a prediction of potential complexity not considering certain natural and anthropogenic disturbances. As was discussed in the reviewer responses, and also added to the discussion of the manuscript, how forest structural complexity will change with climate change depends on the species. Additionally, it depends on the evolutionary history and genetic structure and diversity of the forest communities. Tree species also tend to be plastic, quite adaptable and not shift in range as much as one would think of considering their optimal environmental niche. It seems more likely to see strong shifts in species and functional composition after large-scale disturbances such as insect attacks/forest diebacks, storms or large fires, when forests regrow and assemble in a new way. This is very difficult to predict though, and also depends on the local species and gene pools. Without considering abrupt changes like this, it is still also difficult to predict gradual changes of forest structure with climate change. Species diversity is considered to provide an “insurance” effect, based on the assumption that with a larger species pool it is more likely to have species that can adapt to and be competitive in a new climate or under climate extremes. Therefore, high diversity communities are considered more stable (likely also in structural complexity) and are supposed to adapt better to climate change than low diversity communities. How is this considered in your prediction? In the end, it also really depends on which species might be more vulnerable and die. An experimental study by Bunker et al. (2005) showed that biomass would vary by more than 600% and biological insurance varied by more than 400% among different extinction scenarios, depending on which species die (first). Therefore, I don’t think that one can predict the potential change in structural complexity without historical data (or experiments) showing how specific forest communities and species assemblages react to climate change. I don’t think one can assume the same response (change of SSCI) for different species assemblages with varying evolutionary history globally to a change in precipitation and precipitation seasonality. Trisos et al. (2020) for example show that certain species might experience unprecedented climatic conditions as early as 2030 and some more until 2050 that are outside their

historic climatic niches. How does the prediction of potential SSCI change take into account different species climatic niches, species plasticity and likelihood to migrate or die? You could show predictions of precipitation and/or precipitation seasonality change until 2070, and discuss where we might see positive or negative changes in structural complexity (without estimating the climate effect on SSCI), but the potential “level” for structural complexity to be expected in the future in a region is not just defined by the climate but also by the species pools and evolutionary history. Unless I misunderstand the modeling, I don’t think any of this is taken into account when the authors estimated the climate change effect on SSCI.

Bunker, D. E. (2005). Species Loss and Aboveground Carbon Storage in a Tropical Forest. *Science*, 310(5750), 1029–1031. <https://doi.org/10.1126/science.1117682>

Trisos, C. H., Merow, C., & Pigot, A. L. (2020). The projected timing of abrupt ecological disruption from climate change. *Nature*, 580(7804), 496–501. <https://doi.org/10.1038/s41586-020-2189-9>

- We fully agree with the reviewer that climate change-induced changes in actual forest structural complexity cannot be modelled without taking species- or community-specific responses, evolutionary history and changes in disturbance regimes into account.
- The map in Figure 4c was not intending to show how forest structural complexity will potentially change by 2070, but how the potential for certain levels of structural complexity, i.e. the potentially possible climatic climax of structural complexity, will change relative to our predictions for current climates.
- However, based on the reviewer’s comments, we think that the change map that was shown in figure 4c left too much room for misinterpretations. To avoid misinterpretations of our findings, in e.g. citations, we now follow the reviewer’s suggestion and simply discuss the impacts of predicted changes in annual precipitation and precipitation seasonality on the potential structural complexity.
- We have removed the change map from Figure 4 and now show predicted changes in annual precipitation and precipitation seasonality instead. We decided to place these new figures in the Supplementary Information files (Supplementary Fig. 10) and not in the main manuscript, because they are based on data that was published already.
- We have changed the respective passages in the manuscript accordingly (Line 45-48, 293-317).

Figure SI.10. Relative changes in mean annual precipitation (a) and precipitation seasonality (b) under a RCP8.5 emissions scenario. Projections are based on 17 different climate models that were used in the 5th IPCC report within the frame of the Coupled Model Intercomparisons Project (CMIP5). Maps are based on the WorldClim dataset and show average change across the 17 climate models.

References

Ehbrecht, M., Schall, P., Ammer, C., & Seidel, D. (2017). Quantifying stand structural complexity and its relationship with forest management, tree species diversity and microclimate. *Agricultural and Forest Meteorology*, 242, 1-9.

Ehbrecht, M., Schall, P., Ammer, C., Fischer, M., & Seidel, D. (2019). Effects of structural heterogeneity on the diurnal temperature range in temperate forest ecosystems. *Forest Ecology and Management*, 432, 860-867.

Jucker, T., Hardwick, S. R., Both, S., Elias, D. M., Ewers, R. M., Milodowski, D. T., ... & Coomes, D. A. (2018). Canopy structure and topography jointly constrain the microclimate of human-modified tropical landscapes. *Global change biology*, 24(11), 5243-5258.

Lutz, J. A., Furniss, T. J., Johnson, D. J., Davies, S. J., Allen, D., Alonso, A., ... & Zimmerman, J. (2018). Global importance of large-diameter trees. *Global Ecology and Biogeography*, 27(7), 849-864.

Marra, R. E., Brazee, N. J., & Fraver, S. (2018). Estimating carbon loss due to internal decay in living trees using tomography: implications for forest carbon budgets. *Environmental Research Letters*, 13(10), 105004.

Thonicke, K., Billing, M., Bloh, W., Sakschewski, B., Niinemets, Ü., Peñuelas, J., ... Walz, A. (2020). Simulating functional diversity of European natural forests along climatic gradients. *Journal of Biogeography*, 47(5), 1069–1085. <https://doi.org/10.1111/jbi.13809>

Zemp, D. C., Ehbrecht, M., Seidel, D., Ammer, C., Craven, D., Erkelenz, J., ... & Kreft, H. (2019). Mixed-species tree plantings enhance structural complexity in oil palm plantations. *Agriculture, Ecosystems & Environment*, 283, 106564.

Zenner, E. K., & Hibbs, D. E. (2000). A new method for modeling the heterogeneity of forest structure. *Forest ecology and management*, 129(1-3), 75-87.

Reviewer Comments, third round -

Reviewer #2 (Remarks to the Author):

The authors did a fantastic job in revising the manuscript and discussing my comments and concerns. I would now recommend this manuscript for publication, without further concerns. I do not have any additional corrections, but would like to ask to forward my comments below to the authors.

Thank you very much for making the code available and for planning on publishing the final maps as GeoTiff. This is extremely helpful and can potentially allow future comparisons with other diversity or complexity measures.

I agree that the big trees also have a large influence on total structural complexity of the forest, and I'm therefore fine with the suggested definition.

It is true that microclimate in a forest is a lot driven by the forest and its structure, which again also varies with topography. I therefore agree that including microclimate might confuse the figure, with too many complex interactions between topography, forest structure, and climate.

Thank you for considering the work of Thonicke and colleagues. This is extremely interesting, but I also agree that investigating the relationships between structural complexity and functional diversity is outside the scope of this paper. This has potential for future work. The relationship between structural complexity and functional richness is interesting. I only partially agree with the interpretation though. Functional richness also saturates at high levels of species richness due to functional redundancy of species, and is sometimes higher at lower levels of species richness due to intra-specific diversity. It might also be an effect of how the indices are designed, and the functional traits considered. Functional richness reacts very strongly to extreme values and the functional space might be extended by a few extreme values. These can be related to leaf traits such as LMA that might vary much more than for example tree height. There could also be an effect of scale and the modeling you applied. Is the spatial grain and extent comparable? Can the model predict the extremes of structural complexity which might happen locally? Functional evenness is less scale-dependent, and there seems to be a linear relationship, which makes sense to me. As you stated earlier, structural complexity can be influenced by a few big trees and thus potentially an uneven distribution of functional traits. I think higher structural complexity should be characterized by higher functional divergence and lower evenness. I'm not sure why there is no relationship with divergence, maybe the divergence index is not the best one or it is again driven by certain foliar functional traits that vary differently than structural traits.

Thank you for applying those changes and removing the map, which might lead to too much misinterpretation and potentially a misleading message. With that my concerns are addressed satisfactorily, and I believe that the manuscript has been much improved. I'd like to congratulate the authors for a great paper.

Reviewer #2 (Remarks to the Author):

The authors did a fantastic job in revising the manuscript and discussing my comments and concerns. I would now recommend this manuscript for publication, without further concerns. I do not have any additional corrections, but would like to ask to forward my comments below to the authors.

Thank you very much for making the code available and for planning on publishing the final maps as GeoTiff. This is extremely helpful and can potentially allow future comparisons with other diversity or complexity measures.

I agree that the big trees also have a large influence on total structural complexity of the forest, and I'm therefore fine with the suggested definition.

It is true that microclimate in a forest is a lot driven by the forest and its structure, which again also varies with topography. I therefore agree that including microclimate might confuse the figure, with too many complex interactions between topography, forest structure, and climate.

- We thank the reviewer for the very thorough review of our manuscript. The detailed comments and suggestions helped us a lot to improve the manuscript and we really appreciate the positive feedback on our revisions.

Thank you for considering the work of Thonicke and colleagues. This is extremely interesting, but I also agree that investigating the relationships between structural complexity and functional diversity is outside the scope of this paper. This has potential for future work. The relationship between structural complexity and functional richness is interesting. I only partially agree with the interpretation though. Functional richness also saturates at high levels of species richness due to functional redundancy of species, and is sometimes higher at lower levels of species richness due to intra-specific diversity. It might also be an effect of how the indices are designed, and the functional traits considered. Functional richness reacts very strongly to extreme values and the functional space might be extended by a few extreme values. These can be related to leaf traits such as LMA that might vary much more than for example tree height. There could also be an effect of scale and the modeling you applied. Is the spatial grain and extent comparable? Can the model predict the extremes of structural complexity which might happen locally?

- Thonicke et al.'s map has a coarser spatial resolution than our map of potential structural complexity. Thus, the spatial grain is not fully comparable. We therefore decided to approach a comparison between functional diversity and structural complexity by using a sampling grid with a distance of 100 km between sample points to reduce spatial autocorrelation and to avoid sampling values from Thonicke et al.'s map twice.
- Our model predicts the potential structural complexity using mean annual precipitation and precipitation seasonality as predictor variables. Thus, it can predict local extremes based on local extremes in annual precipitation and precipitation seasonality. As pointed out in the discussion of the manuscript, small-scale variability, and thus small-scale extremes due to differences in soil conditions need to be taken into account for more accurate local-scale predictions.
- Focusing future studies on local-scale variability of forest structural complexity could help to refine the world maps presented in this study.
- We fully agree with the reviewer that functional richness may saturate at high levels of species richness due to functional redundancy. We assume that this could explain a

saturation of forest structural complexity at high levels of functional richness due to a functional redundancy of some species in relation to structural traits, such as tree heights and crown architectures.

Functional evenness is less scale-dependent, and there seems to be a linear relationship, which makes sense to me. As you stated earlier, structural complexity can be influenced by a few big trees and thus potentially an uneven distribution of functional traits. I think higher structural complexity should be characterized by higher functional divergence and lower evenness. I'm not sure why there is no relationship with divergence, maybe the divergence index is not the best one or it is again driven by certain foliar functional traits that vary differently than structural traits.

- We agree with the reviewer. It could be that functional divergence measure here is driven by certain foliar traits that are not as relevant to forest structural complexity as other structural traits, such as tree heights.
- Recent advances in mapping functional diversity across spatial scales could be a starting point to address relationships between functional diversity and forest structural complexity in more detail, for example:
 - Asner et al. (2017) Airborne laser-guided imaging spectroscopy to map forest trait diversity and guide conservation. *Science*, 355(6323), 385-389, DOI: [10.1126/science.aaj1987](https://doi.org/10.1126/science.aaj1987),
 - Schneider et al. (2017), Mapping functional diversity from remotely sensed morphological and physiological forest traits. *Nature Communications*, 8(1), 1-12, doi.org/10.1038/s41467-017-01530-3,
 - Thonicke et al. (2020), Simulating functional diversity of European natural forests along climatic gradients, *Journal of Biogeography*, 47(5), 1069-1085, doi.org/10.1111/jbi.13809
- We have complemented the discussion (line 225-230) to highlight the need and potential for future research in this direction. Exploring the relationships between functional diversity and structural complexity could be a key to unravel the mechanistic underpinnings of climate-structure relationships in more detail.

Thank you for applying those changes and removing the map, which might lead to too much misinterpretation and potentially a misleading message. With that my concerns are addressed satisfactorily, and I believe that the manuscript has been much improved. I'd like to congratulate the authors for a great paper.

- Ultimately, we agree with the reviewer. The removed map could have led to misinterpretation and misleading messages. We also think that the manuscript rather gained clarity without the change map.